# `PLoRA`: Efficient Concurrent LoRA Training for Large Language Models

**Minghao Yan** [† * 1]   **Zhuang Wang** [* 2]   **Zhen Jia** [2]   **Shivaram Venkataraman** [1]   **Yida Wang** [2]

## Abstract

Low-Rank Adaptation (LoRA) has gained popularity as a fine-tuning approach for Large Language Models (LLMs) due to its low resource requirements and good performance. While numerous studies have investigated ways to improve LoRA serving efficiency by serving multiple LoRAs concurrently, existing methods assume that a wide range of LoRA adapters are available for serving. In our work, we conduct extensive empirical studies to show that current LoRA training paradigms do not efficiently utilize hardware resources and incur high overhead to obtain a performant LoRA adapter. Leveraging these insights, we propose `PLoRA`, which automatically orchestrates concurrent LoRA fine-tuning jobs under given hardware and model constraints and develops performant kernels to improve training efficiency. Across a range of LLMs and LoRA configurations, `PLoRA` improves training throughput by up to $12.8\times$ and reduces the overall fine-tuning makespan by up to $7.52\times$ compared to existing approaches.

## 1. Introduction

Large Language Models (LLMs) have become the backbone of numerous modern AI applications, spanning natural language understanding, code generation, multimodal reasoning, and specialized domains such as healthcare and finance (Guo et al., 2025; Grattafiori et al., 2024; Yang et al., 2024; Abdin et al., 2024; Wang et al., 2023; Team et al., 2024). The paradigm of pretraining followed by fine-tuning has enabled models to achieve state-of-the-art performance when adapted to specific tasks (Ouyang et al., 2022). However, fine-tuning large models for multiple tasks or user-specific applications is challenging due to the high computational cost of training and serving numerous fine-tuned vari-

ants. To address this, parameter-efficient fine-tuning (PEFT) techniques such as Low-Rank Adaptation (LoRA) (Hu et al., 2021a; Gunter et al., 2024; Ding et al., 2023) have emerged as scalable alternatives to full fine-tuning. LoRA significantly reduces the number of trainable parameters by introducing low-rank decomposition matrices into Transformer layers, allowing for specialization while keeping the pretrained model weights frozen. Due to the popularity of this versatile deployment approach, several systems have been developed to serve multiple LoRA adapters concurrently (Sheng et al., 2023; Chen et al., 2024).

While LoRA serving has received significant attention, existing systems largely assume that high-quality LoRA adapters are already available. In practice, however, users rarely train a single adapter in isolation (Schulman and Lab, 2025). Instead, LoRA fine-tuning commonly involves *many independent training jobs*, arising from hyperparameter exploration (Fomenko et al., 2024), adaptation to multiple downstream tasks (Li et al., 2024a;b), or specialization to different data domains (Gunter et al., 2024). This paper focuses on a complementary but underexplored question: *how can we efficiently train many LoRA adapters at scale?*

In addition to standard optimization parameters such as learning rate and batch size, LoRA configurations vary in architectural choices, including adapter rank and scaling factor (Figure 1), which jointly affect model quality, memory footprint, and computational cost (Hu et al., 2021a; Fomenko et al., 2024). In practice, an enterprise that offers fine-tuning-as-a-service on a shared base model may simultaneously adapt the model for customer support, code assistance, document analysis, or safety alignment (Zhao et al., 2024; Gunter et al., 2024). These adaptations often rely on different datasets, optimization objectives, and loss functions, and may require distinct LoRA configurations to achieve satisfactory performance.

Despite this, current LoRA fine-tuning pipelines execute each configuration independently. Our empirical analysis reveals that this execution model is highly inefficient. Individual LoRA fine-tuning jobs are lightweight: they typically use small batch sizes, involve low-rank matrix operations, and update only a tiny fraction of model parameters. Consequently, they underutilize modern GPUs, exhibiting SM occupancy as low as $16.7\%$ and memory utilization below

---

[*]Equal contribution . [†]Work partially done during an internship at Amazon. [1]University of Wisconsin-Madison [2]Amazon Web Services. Correspondence to: Minghao Yan <myan@cs.wisc.edu>.

*Proceedings of the 43rd International Conference on Machine Learning*, Seoul, South Korea. PMLR 306, 2026. Copyright 2026 by the author(s).

55% across a wide range of settings (§2.3). This inefficiency is exacerbated when multiple LoRA jobs are required, leading to long makespans despite abundant hardware resources.

These observations point to a key insight: the dominant bottleneck in LoRA fine-tuning is not a lack of parallel work, but the isolation of many heterogeneous training jobs. Motivated by this, we propose to *pack multiple LoRA adapters into a single fine-tuning run*, allowing them to share hardware resources and execute concurrently, improving LoRA training efficiency.

In this paper, we present PLoRA, an automated system for concurrent LoRA fine-tuning. Given a base model and a set of LoRA configurations, PLoRA coordinates efficient execution through a two-stage design. First, PLoRA uses an offline packing planner to analyze the configurations and groups them into packed jobs that maximize hardware utilization while respecting model and resource constraints. We formulate this planning problem as a Knapsack problem and develop an efficient approximation algorithm with provable performance bounds (§3.2). Second, we design an online execution engine that dynamically deploys these jobs, monitors resource availability, and launches packed LoRA fine-tuning runs accordingly. To support efficient execution, we design specialized GPU kernels for packed LoRA adapters and demonstrate near-linear scaling while concurrently training up to 32 adapters across multiple base models. In summary, we make the following contributions:

- We demonstrate that standard LoRA fine-tuning underutilizes modern GPUs and that this inefficiency is amplified in multi-LoRA workflows.

- We introduce a packing-based framework, PLoRA, for concurrent LoRA training and develop an optimization-based planner that efficiently schedules heterogeneous LoRA configurations.

- We design and implement a high-performance execution engine in PLoRA and demonstrate that PLoRA achieves up to $12.8\times$ throughput improvement and reduces total training makespan by up to $7.52\times$.

## 2. Motivation

### 2.1. Low-Rank Adaptation (LoRA)

Low-Rank Adaptation (LoRA) (Hu et al., 2021a) is a widely adopted efficient fine-tuning technique for Large Language Models (LLMs). Formally, for a weight matrix $W \in \mathbb{R}^{d \times k}$, LoRA adds the weight updates $\Delta W$ as two small matrices $A \in \mathbb{R}^{d \times r}$ and $B \in \mathbb{R}^{r \times k}$, where $r$ is the LoRA rank and much smaller than $d$ and $k$. The additional FLOPs incurred by LoRA are linear to its rank. LoRA updates only $A$ and $B$, thereby significantly reducing computation and storage

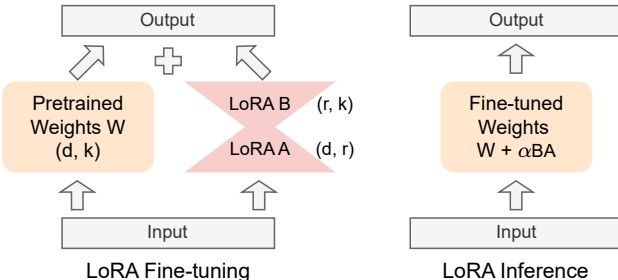

*Figure 1.* This figure demonstrates how LoRA is applied to weight matrices in fine-tuning and inference.

*Table 1.* Hyperparameters for LoRA fine-tuning.

| Hyperparameters | Search range | Meaning |
|---|---|---|
| Learning rate (LR) | 2e-5 $\sim$ 4e-4 | Step size |
| Batch size (BS) | $1 \sim 32$ | Batch size |
| LoRA rank (r) | $8 \sim 128$ | LoRA rank |
| LoRA alpha ($\alpha$) | $r/4 \sim 4r$ | LoRA scaling factor |

costs during fine-tuning.

During inference, LoRA merges the multiplied matrix $\Delta W = B \times A$ into the original weight matrix $W$ with a scaling factor $\alpha$, and the weight matrix becomes $W = W + \alpha \Delta W$, as shown in Figure 1.

### 2.2. LoRA Configuration Space

Modern LoRA workflows involve selecting a set of configuration parameters that govern both optimization behavior and adapter capacity. In addition to hyperparameter tuning, similar configuration choices arise when multiple LoRA adapters are trained for different tasks, domains, or use cases. These include learning rate and batch size (Bergstra et al., 2011; Bergstra and Bengio, 2012), as well as LoRA-specific architectural choices.

In particular, LoRA introduces additional configuration dimensions, most notably the adapter rank $r$ and the scaling factor $\alpha$, which directly control the adapter's expressive capacity and its interaction with the frozen base model. The configuration space explored in this work is summarized in Table 1. Unlike full fine-tuning, where hyperparameters are often transferable across tasks, LoRA performance is highly sensitive to these architectural choices (Schulman and Lab, 2025). Moreover, the required adapter capacity varies substantially across workloads: for example, supervised instruction tuning often benefits from higher ranks than reinforcement-learning-based adaptation (Schulman and Lab, 2025).

As a result, real-world LoRA training pipelines must routinely handle a large and heterogeneous set of configurations. Whether these configurations arise from explicit hyperparameter search or from the need to train multiple adapters

for diverse downstream requirements, they collectively form a large configuration space that must be explored efficiently.

## 2.3. Model Quality versus Hardware Efficiency

The motivation for PLoRA arises from a fundamental tension between the optimization requirements of LoRA fine-tuning and the execution characteristics of modern GPUs.

**LoRA optimization favors small batch sizes.** While large batch sizes are commonly used in pretraining to maximize throughput, prior work (Fomenko et al., 2024) and our study (Appendix E.1) both observe a degradation in LoRA performance as batch size increases due to different optimization dynamics (Shuttleworth et al., 2024).

**Small batch sizes conflict with GPU efficiency.** Modern accelerators such as NVIDIA A100 and H100 GPUs are designed to achieve peak performance through high arithmetic intensity and massive parallelism. However, the combination of small batch sizes and low-rank adapter updates yields insufficient computational workload to occupy GPU resources fully. In practice, this results in substantial underutilization of compute and memory during LoRA fine-tuning.

**Multiple LoRA jobs amplify inefficiency.** In practice, LoRA fine-tuning is rarely performed for a single configuration (Zhao et al., 2024). Users often need to train many LoRA adapters, whether to explore different configurations, support multiple downstream tasks, or adapt the same base model across diverse domains. When each LoRA adapter is fine-tuned in isolation, the inefficiency of a single run compounds across all jobs. As a result, the dominant bottleneck is not a lack of work or model capacity, but the inability of individual LoRA fine-tuning jobs to utilize modern hardware when executed independently efficiently.

**Low and invariant SM occupancy.** We profile single-adapter LoRA fine-tuning on an NVIDIA A100 GPU using QWen-2.5-7B with Unsloth (Han et al., 2023). Across batch sizes from 1 to 16 and LoRA ranks from 8 to 128, SM occupancy remains consistently low at approximately $16.7\%$ for both base model and LoRA kernels. This indicates that LoRA's matrix operations lack sufficient arithmetic intensity and data reuse to fully exploit GPU parallelism.

**GPU memory underutilization.** During LoRA fine-tuning, memory usage is dominated by the frozen base model, while the trainable LoRA adapters occupy only a small fraction of GPU memory. Combined with small batch sizes, this leaves substantial memory capacity unused throughout training.

**Implication.** These results indicate that LoRA fine-tuning is constrained not by model quality or data availability, but by inefficient utilization of modern GPU resources. Since practical deployments often require training many LoRA

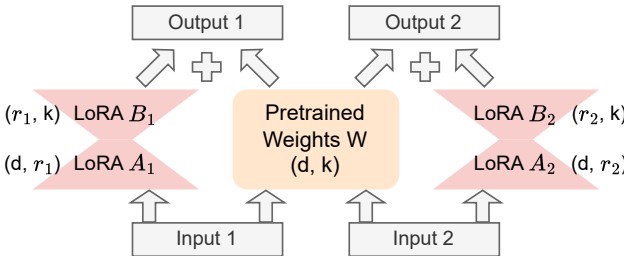

*Figure 2.* An illustration of packed LoRA fine-tuning with two LoRA adapters sharing a frozen base model.

adapters, the inefficiency of a single fine-tuning run compounds across multiple jobs. This observation suggests that improving LoRA training efficiency requires rethinking the execution model. Rather than optimizing individual runs in isolation, multiple LoRA configurations should be executed concurrently on shared hardware to amortize better underutilized compute and memory resources.

## 2.4. Our Proposal: Packed LoRA Fine-tuning

To address this inefficiency, we propose *packed LoRA fine-tuning* (PLoRA), which concurrently fine-tunes multiple LoRA configurations on a shared frozen base model to maximize hardware utilization.

**Packing LoRA adapters is feasible.** Each LoRA configuration corresponds to a distinct LoRA adapter, while the base model remains identical across all configurations and is frozen during fine-tuning. This enables multiple adapters to be packed into a single fine-tuning job without increasing base model memory. When combined with tensor parallelism or FSDP (Zhao et al., 2023), the aggregate memory capacity allows a large number of LoRA adapters to be packed without incurring out-of-memory (OOM) errors.

**Packed LoRA fine-tuning workflow.** Contrary to standard LoRA fine-tuning (Figure 1), packed LoRA fine-tuning processes a set of inputs, one per adapter, where each adapter maintains its own parameters, optimizer state, training data, and loss function, enabling adapters targeting different downstream tasks or objectives to be trained concurrently, as shown in Figure 2. All inputs are forwarded through the shared base model, while each LoRA adapter applies its own low-rank update. The computation performed for each adapter is identical to standard LoRA fine-tuning, but sharing the base model enables higher hardware utilization.

## 2.5. Challenges

While packed LoRA fine-tuning improves hardware utilization, it introduces new system-level challenges that do not arise in standard single-adapter fine-tuning.

**Efficient computation of packed LoRA adapters.** A

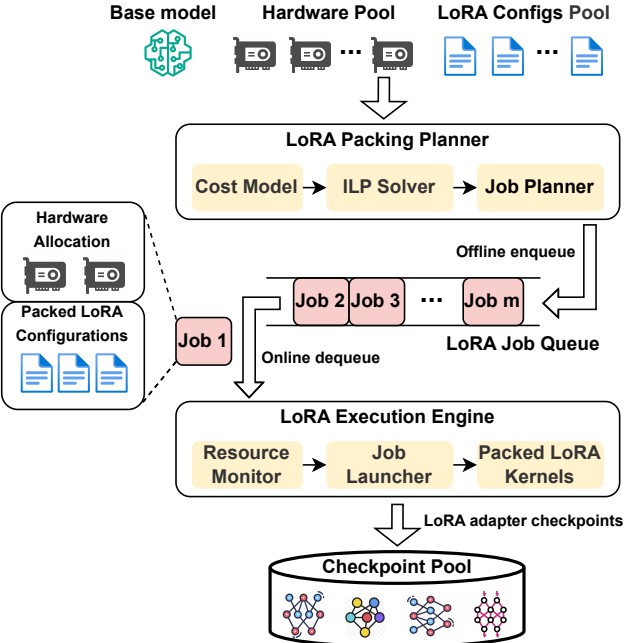

*Figure 3.* The system architecture of PLoRA.

PLoRA operates in two phases: *offline configuration planning* and *online execution*. A *fine-tuning job* is defined as fine-tuning multiple packed LoRA adapters on a shared base model. In the offline phase, the Packing Planner explores packed configurations using a cost model that estimates memory usage and throughput from the first few iterations (10 in our testbed). The Job Planner then determines packing strategies, allocates hardware resources, and enqueues planned jobs.

In the online phase, the Execution Engine dynamically dequeues jobs based on available hardware. Its Job Launcher configures parallelism strategies and deploys packed jobs, while the Resource Monitor monitors resource availability. PLoRA can run multiple jobs concurrently as long as the resources are available. Customized Packed LoRA Kernels improve GPU utilization during both forward and backward propagation (Appendix C).

Upon job completion, each adapter is stored in the Checkpoint Pool, and the released hardware resources are returned to the pool. The Resource Monitor then triggers the execution of the next queued jobs, ensuring continuous and efficient use of hardware.

### 3.2. Scheduling of Packed LoRA Fine-tuning

This section describes how to schedule packed LoRA configurations for concurrent LoRA fine-tuning. We first formalize the optimization problem by jointly considering LoRA configuration packing and hardware allocation for fine-tuning jobs. Since the problem is NP-hard, we develop an approximate algorithm and analyze its performance.

The optimization goal is to minimize the *makespan* of training time for all configurations in the given search space on the specified hardware. We observe that the completion time of a LoRA fine-tuning job is mainly affected by two factors:

- **The packed LoRA configurations**, which determine the set of LoRA adapters fine-tuned in a job.

- **The degree of parallelism**, which determines the number of GPUs used for each fine-tuning job.

$$\max \sum_{j=1}^{m} \frac{\sum_{k=1}^{|K|} \mathcal{H}_{j,k} * r_k}{T(\mathcal{H}_{j,k}, d_j)}, \tag{1}$$

$$\text{s.t.} \quad M_{\text{base}} + \sum_{k=1}^{|K|} \mathcal{H}_{j,k} * M_{\text{lora},k} \leq C * M_{\text{gpu}} * d_j, \tag{2}$$

$$\forall 1 \leq j \leq m$$

$$\Sigma_j d_j \leq G, \quad 1 \leq j \leq m \tag{3}$$

$$1 \leq d_j \leq G, \quad d_j \in \{2^i \mid i \in \mathbb{N}\} \tag{4}$$

$$m \geq 1, \quad m \in \mathbb{Z} \tag{5}$$

packed fine-tuning job consists of a shared base model and multiple LoRA adapters with distinct parameters and inputs. While iterating over adapters and inputs sequentially can still benefit from amortized memory costs, it results in low utilization during both forward and backward passes due to small LoRA ranks and limited arithmetic intensity.

**Resource-aware packing and scheduling.** Even with optimized kernels, hardware resources cannot simultaneously accommodate a large number of configurations. Maximizing throughput requires jointly determining how LoRA adapters are packed into fine-tuning jobs and how GPU resources are allocated across jobs, while avoiding OOM errors. Optimizing packing or allocation in isolation is insufficient as we show in §4.5.

## 3. PLoRA: Enable Concurrent LoRA Fine-tuning

### 3.1. System Overview

We propose PLoRA, a system for efficient LoRA training via packed fine-tuning. Given a base model, a hardware pool, and a configuration search space, PLoRA maximizes tuning throughput by packing LoRA configurations to utilize hardware resources fully. Figure 3 illustrates PLoRA, which has two main components: 1) a *LoRA Execution Engine* that launches packed fine-tuning jobs with optimized Packed LoRA Kernels (§C), and 2) a *LoRA Packing Planner* that schedules configurations by jointly optimizing packing and hardware allocation (§3.2).

The optimization problem aims to minimize the makespan $t_{opt}$. The detailed formulation is provided in Appendix A. This optimization problem is NP-complete as it can be viewed as a variant of a 0-1 knapsack problem (Karp, 2009). Our goal is to solve the makespan problem (Eq. 8). We approximate this by maximizing instantaneous throughput (Eq. 1), which we solve using DTM (Alg. 1) and prove to be near-optimal (Appendix D). In addition, our optimization formulation adds a new layer of complexity, as each job must first decide which LoRA configurations ($\mathcal{H}_{jk}$) to train and determine the associated degree of parallelism.

Since the LoRA FLOP (floating-point operations of adapters, excluding the base model) is fixed given a search space, minimizing the makespan ($t_{opt}$) is equivalent to maximizing the average LoRA FLOP over this time. Thus, the optimization problem can be rewritten in throughput form as $\max \frac{\sum_{k=1}^{|K|} FLOP_k}{t_{opt}}$, where $FLOP_k$ denotes the FLOP of configuration $k$. Because solving this problem exactly is intractable, we next develop an approximation algorithm to address it.

$$F(D, K) = \frac{\sum_{k=1}^{|K|} \mathcal{H}_k * r_k}{T(\mathcal{H}_k, D)}, \quad (6)$$

$$\text{s.t.} \quad M_{\text{base}} + \sum_{k=1}^{|K|} \mathcal{H}_k * M_{\text{lora},k} \leq C * M_{\text{gpu}} * D, \quad (7)$$

Computing an optimal job schedule for average throughput is challenging, so we instead maximize *instantaneous job-level throughput*. This leads to a scheduling algorithm with provable bounds on the optimality gap.

**Maximizing fine-tuning job throughput.** Given $G$ GPUs and a set of LoRA configurations $K$, we approximate minimizing makespan by maximizing throughput (Expression 1).

Note that we use LoRA rank in Eq (1) instead of LoRA FLOP by leveraging the linear scaling property of LoRA FLOP in rank (refer to §2.1). Here $r_k$ is the rank of configuration $k$, $m$ the number of concurrent jobs, $d_j$ the parallelism degree of job $j$, $M_{\text{base}}$ the base model memory, $M_{\text{lora},k}$ the LoRA memory, $M_{\text{gpu}}$ the GPU capacity, and $C \in (0, 1]$ a load factor. Constraints 2–5 enforce memory, GPU, and parameter ranges.

**ILP Solvable with determined parallelism degree.** We note that the problem is nonconvex because $T(\mathcal{H}_{j,k}, d_j)$ depends on $d_j$. However, since $d_j$ takes power-of-two values, we can enumerate the denominator. We thus solve a restricted ILP where parallelism is fixed at $D \leq G$. Here $\mathcal{H}_k$ indicates whether configuration $k$ is selected. We solve $F(D, K)$ with a DTM algorithm (Alg. 1): for each $D$, the solver optimizes $F(D, K)$, then applies ILP to the remaining subproblems. The algorithm terminates when no GPUs

---

**Algorithm 1** Decomposed Throughput Maximization (DTM)

---

**input** Number of GPUs $G$, LoRA configuration space $K$
**output** Scheduling policy $P$ (packed LoRA configurations).

1: **function** DTMHelper$g, P_{tmp}, K, P$
2:     **if** $g \leq 0$ **or** $K = \emptyset$ **then**
3:         $P \leftarrow P \cup P_{tmp}$
4:         **return**
5:     **end if**
6:     $g' \leftarrow 2^{\lfloor \log_2 g \rfloor}$
7:     **for** $d \in \{g', g'/2, \ldots, 1\}$ **do**
8:         Call Gurobi ILP solver
9:         $P_{new}, K_{used} \leftarrow F(d, K)$
10:         **call** DTMHelper($g - d, P_{tmp} \cup P_{new}, K - K_{used}$)
11:     **end for**
12: **end function**
13: **function** DTM$G, K$
14:     $P \leftarrow \emptyset$
15:     **call** DTMHelper($G, \emptyset, K, P$)
16:     **return** $\arg\min\{T(p) \mid p \in P\}$
17: **end function**

---

remain or all configurations are scheduled. Among all candidate schedules $P$, the one with the minimum makespan is selected.

**The job planner.** Algorithm 1 focuses on finding the best packing of LoRA configurations on the available GPUs to maximize concurrent throughput. It determines which configurations to train together and to what degree of parallelism. The job scheduling algorithm then takes these packed groups and decides the order in which they should run on the hardware, producing a complete job queue. Together, these two algorithms first determine the most efficient packing of configurations (Algorithm 1) and then schedule those packs over time (job scheduler) (Algorithm 2). This overall process approximately solves Problem (1), which is equivalent to minimizing the makespan, by keeping hardware as fully utilized as possible throughout execution.

The core principle of the job planner is to schedule packed LoRA fine-tuning jobs to maximize concurrent throughput whenever hardware resources are available. If there are available GPU resources for job scheduling (Line 4), it invokes DTM() introduced in Algorithm 1 to find the best set of packed LoRA fine-tuning jobs for these available resources (Line 5) and updates the remaining LoRA configurations (Line 7). The job planner also adds the set of jobs to the LoRA job queue (Line 8). It then predicts the next job completion event with the cost model and updates the number of available GPUs for the next round of job planning.

**Algorithm 2** The Job Planner

**input** Number of GPUs $G$, LoRA configuration space $K$
**output** LoRA job queue $Q$

```
 1: Q ← []
 2: g_avail ← G
 3: while K ≠ ∅ do
 4:    if g_avail > 0 then
 5:        P ← DTM(g_avail, K)
 6:        for all p ∈ P do
 7:            K ← K \ p.configs
 8:        end for
 9:        Q.append(P)
10:    end if
11:    Predict next job completion event
12:    Update g_avail
13: end while
14: return Q
```

**Computation time of the job planner.** The job planner's runtime is negligible, especially given that this is for offline scheduling. Since solving each optimization instance takes less than a second and all sub-branches can be performed in parallel, we observe that the computation time of Algorithm 1 is within 10 minutes in our evaluation with 120 configurations on 8 GPUs(§4.2), less than 2.5% of the overall duration. In Appendix D, we prove the near-optimality of the scheduling algorithm.

# 4. Evaluation

In this section, we will demonstrate the effectiveness of PLoRA for large-scale LoRA fine-tuning. Specifically, we will address the following questions: 1. Can PLoRA reduce the makespan of large-scale LoRA fine-tuning? (§4.2) 2. Does PLoRA find better LoRA adapters that improve model quality? (§4.4) We also conduct detailed ablation studies to evaluate each component's performance.

## 4.1. Experiment Setup

**Testbed.** We run experiments on a g5 and a p4d.24xlarge instance on Amazon EC2. The P4d instance has 8 A100 GPUs (40 GB each) connected via NVLink for GPU-to-GPU communication. The G5 instance has 8 A10 GPUs (24 GB each) connected via PCIe Gen4 for GPU-to-GPU communication.

**Models and tasks.** We conduct experiments on the Qwen 2.5 model family, one of the frontier open-weight model families that provides the most complete selection of sizes, including $3B$, $7B$, $14B$, and $32B$, as well as on LLAMA-3.2-3B and LLAMA-3.1-8B. We perform our evaluation in a zero-shot setting, following the prompting template in prior

work (Zhao et al., 2024). We use four downstream tasks, GSM8K (Cobbe et al., 2021), mrpc (Wang et al., 2018), cola (Wang et al., 2018), and wnli (Wang et al., 2018) and set sequence length to 1024.

**LoRA configuration selection.** As discussed in §2.2, the search space in our evaluations consists of four knobs: learning rate, batch size, LoRA rank, and LoRA scaling factor $\alpha$. Their ranges are listed in Table 1. We select 120 LoRA configurations for the experiments based on a detailed empirical sensitivity analysis (Appendix E.1).

**Baselines.** We compare PLoRA with existing approaches for LoRA fine-tuning, in which each LoRA fine-tuning job only evaluates one LoRA adapter. We consider two strategies for sequential approaches: *Min GPU*, which uses the minimum set of hardware that satisfies the memory constraints for each LoRA fine-tuning job and launches parallel jobs to fill all GPUs; and *Max GPU*, which uses the maximum number of devices within a GPU instance for each LoRA fine-tuning job, i.e., setting TP degree to 8 in our testbed. In the Appendix §B, we show how to deploy PLoRA on FSDP and pipeline parallel (Narayanan et al., 2019) setups.

**Metrics.** We use makespan to evaluate the end-to-end performance of LoRA fine-tuning across all LoRA configurations in the search space. We use throughput to evaluate the performance of LoRA fine-tuning jobs and packed LoRA kernels. We report the zero-shot accuracy on the downstream tasks.

**Implementation.** We implement a prototype of PLoRA atop torchtune (torchtune, 2024). Our implementation contains around 5000 lines of Python code and around 800 lines of CUDA code for customized packed LoRA kernels. We use cvxpy (Diamond and Boyd, 2016) to implement our optimization module and built upon the PyTorch DTensor primitive to customize LoRA tensor parallel sharding strategies for efficient fine-tuning with tensor parallelism.

## 4.2. Makespan Evaluation

We evaluate PLoRA 's makespan improvement relative to two baselines across 120 LoRA configurations. The base models are from the QWen-2.5 family. A single GPU can fit the 3B and 7B models, while the 14B model requires two GPUs and the 32B model requires four GPUs. Thus, for the Min GPU, we use this TP size and start concurrent jobs to fully occupy all available GPUs. For Max GPU, we use all eight GPUs per job, allowing only one job at a time.

Figure 4a shows the makespan normalized to Min GPU. The performance of Max GPU is much worse than that of Min GPU due to even lower hardware utilization. On the contrary, PLoRA reduces the makespan by $6.51\times$ and $6.33\times$ on 14B and 32B, respectively, thanks to packed LoRA fine-tuning. PLoRA also achieves $7.08\times$ and $6.52\times$ reductions

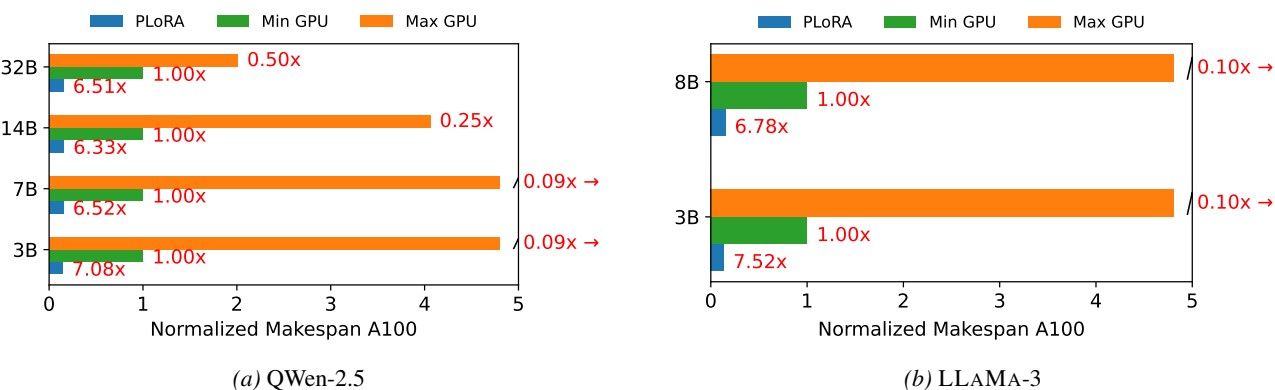

*(a)* QWen-2.5          *(b)* LLaMa-3

*Figure 4.* The makespan of LoRA fine-tuning with different methods on A100 GPUs. The makespan is normalized to the performance of the Min GPU.

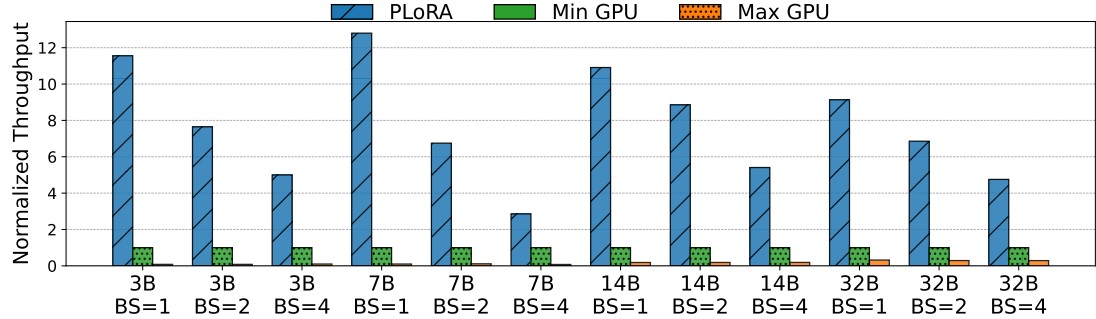

*Figure 5.* LoRA fine-tuning job throughput for various QWen-2.5 model sizes and batch sizes (BS) on A100 GPUs. The performance is normalized to Min GPU.

in makespan on the 3B and 7B models, respectively.

We also evaluate LLaMa-3.2-3B and LLaMa-3.1-8B and observe similar improvements in makespan. Min GPU runs each LoRA fine-tuning job on a single GPU and launches 8 concurrent jobs. Max GPU still has the worst makespan, as shown in Figure 4b. PLoRA achieves $7.52\times$ speedups over Min GPU for LLaMa-3.2-3B and $6.78\times$ speedup for LLaMa-3.1-8B.

### 4.3. Job-level Throughput Evaluation

We study the benefits of packing by measuring the throughput of packed LoRA fine-tuning jobs compared to our baselines. We run PLoRA with different base models and batch sizes on A100 GPUs. We fix LoRA rank to be 32, and other settings of PLoRA, Min GPU, and Max GPU are the same as those in §4.2. We show the job throughput on QWen-2.5 models in Figure 5. We observe similar trends in the LLaMa-3 models.

For a batch size of 1, PLoRA achieves up to $12.8\times$ speedup across the tested models. When we increase the batch size, the performance gain diminishes since the Min GPU strategy can better utilize the hardware. However, the table shows that we still achieve a significant throughput improvement

for a batch size of 4. Further increasing batch sizes harms model quality, as shown in the appendix §E.1 as well as prior work (Fomenko et al., 2024; Schulman and Lab, 2025).

### 4.4. Model Quality with PLoRA

In this section, we evaluate the quality of the best LoRA adapter found by PLoRA in the given search space, which includes 120 LoRA configurations. Four base models, QWen-2.5-3B, QWen-2.5-7B, LLaMa-3.2-3B, and LLaMa-3.1-8B are fine-tuned with LoRA on four downstream tasks.

The model quality results are shown in Table 2. Each cell reports four numbers: (1) the base model without LoRA, (2) the LoRA adapter fine-tuned with the default configurations from Unsloth (Han et al., 2023), a popular LoRA framework, (3) the best LoRA adapter found in our search space, and (4) the quality improvement (in red) of the best configuration over the default one. The results show that default LoRA configurations already improve quality over the base model on downstream tasks. However, they do not fully exploit LoRA's potential. After searching 120 configurations with PLoRA, the best LoRA adapters outperform the default configuration by up to $23.4\%$, with consistent gains across different model families.

*Table 2.* Model quality comparison of different models. The numbers in each cell are the base model quality, the default LoRA configuration, the best LoRA configuration, and the quality improvement over the default configuration.

|  | QWen-2.5-3B | QWen-2.5-7B | LLaMa-3.2-3B | LLaMa-3.1-8B |
|---|---|---|---|---|
| mrpc | 62.4 / 62.6 / 67.6 +5.0% | 64.1 / 64.7 / 70.0 +5.3% | 70.3 / 77.4 / 80.6 +3.2% | 71.3 / 80.3 / 84.5 +4.2% |
| cola | 48.8 / 53.8 / 77.2 +23.4% | 62.7 / 68.4 / 80.2 +11.8% | 69.9 / 71.8 / 77.3 +5.5% | 71.9 / 73.8 / 80.0 +6.2% |
| wnli | 53.5 / 66.2 / 73.4 +7.2% | 78.8 / 80.1 / 84.5 +4.4% | 46.4 / 61.9 / 64.8 +2.9% | 54.9 / 67.6 / 73.2 +5.6% |
| gsm8k | 61.2 / 64.8 / 74.6 +9.8% | 70.8 / 72.1 / 79.8 +7.7% | 60.4 / 63.3 / 71.3 +8.0% | 69.6 / 70.5 / 78.0 +7.5% |

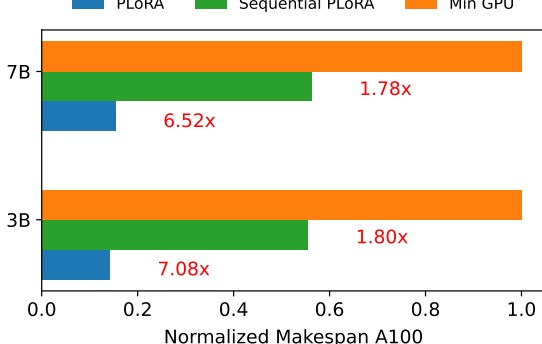

*Figure 6.* This figure shows the breakdown of name's speedup on A100 GPUs. Sequential PLoRA represents the speedup obtained via leveraging name's Packing Planner, but performs vanilla sequential LoRA training without name's Execution Engine.

### 4.5. Speedup Breakdown

PLoRA 's performance gains mainly come from two components: optimized GPU kernels for efficient packed LoRA computations and near-optimal scheduling for packing LoRA configurations. This ablation study breaks down how each component contributes to the overall reduction in makespan. We compare the makespan of end-to-end LoRA fine-tuning with Min GPU, using PLoRA for job planning but executing LoRA sequentially (Sequential PLoRA), and PLoRA. The results, normalized to the Min GPU, are shown in Figure 6. We use QWen-2.5-3B and QWen-2.5-7B as base models, and a search space of 120 LoRA configurations. Sequential PLoRA reduces the makespan by around $1.8\times$ for both models by amortizing the base-model computation. The optimized GPU kernels further reduce the makespan by up to $3.93\times$, demonstrating that both components contribute significantly to PLoRA 's performance.

## 5. Related works

**LoRA.** Prior work has extensively studied efficient *LoRA serving*. DLoRA (Wu et al., 2024), Punica (Chen et al., 2024), and SLoRA (Sheng et al., 2023) develop scheduling policies, optimized GPU kernels, and memory management techniques to support the concurrent serving of many LoRA adapters. These systems, however, focus exclusively on inference-time efficiency and assume that all LoRA adapters have already been trained.

On the training side, mLoRA (Ye et al., 2023) proposes a pipeline-parallel strategy to scale LoRA fine-tuning across devices. In contrast, PLoRA targets the inefficiency in training *multiple independent LoRA adapters*. Rather than optimizing a single training run, PLoRA improves end-to-end system efficiency by packing and executing multiple LoRA adapters concurrently, enabling large-scale multi-job LoRA workflows.

PLoRA can be directly applied to LoRA variants such as QLoRA (Dettmers et al., 2023) (Appendix E.3) and DoRA (Liu et al., 2024), as they only change the base-model weights and do not affect LoRA computations. PLoRA can be used on AdaLoRA (Zhang et al., 2023) by specifying different ranks for different layers.

**Job scheduling.** Makespan minimization is an extensively studied topic in generalized cluster job scheduling (Narayanan et al., 2020; Xiao et al., 2018; Hu et al., 2021b; Grandl et al., 2014; Ghirardi and Potts, 2005). These prior works on generalized cluster scheduling assume jobs are predefined (Grandl et al., 2016a; Gu et al., 2019; Mahajan et al., 2020; Chaudhary et al., 2020; Grandl et al., 2016b). However, in LoRA fine-tuning, name's optimization module also determines how LoRA configurations are packed into each job and the degree of parallelism for each job. We leverage additional information about LoRA configurations and hardware resources to optimize job scheduling and joint allocation of GPU resources.

## 6. Conclusion

This paper presents PLoRA, a system that enables efficient *concurrent* training of multiple LoRA adapters on modern GPU hardware. Motivated by the observation that practical LoRA workflows often require training many heterogeneous adapters across setups and domains, we identify hardware underutilization as a fundamental bottleneck in existing fine-tuning pipelines. To address this challenge, we design a LoRA packing planner and an optimized execution engine that jointly enable packed LoRA fine-tuning within a single run. By sharing a frozen base model and efficiently scheduling adapter execution, PLoRA substantially improves hardware utilization without compromising model quality. Across a wide range of models and configurations, PLoRA improves fine-tuning throughput by up to $12.8\times$ compared to conventional isolated training, and reduces end-to-end makespan by up to $7.52\times$.

## Acknowledgment

We gratefully acknowledge the support of the NSF Diamond project OAC-2311767 (Democratizing Large Neural Network Model Training for Science).

## Impact Statement

This paper presents work whose goal is to advance the field of Machine Learning and improve system efficiency. There are many potential societal consequences of our work, none of which we feel must be specifically highlighted here.

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

# A. Optimization formulation

The optimization problem can be formulated as follows:

$$\min \quad t_{opt} \tag{8}$$

$$\text{s.t.} \quad t_{opt} \geq s_j + T(\mathcal{H}_{j,k}, d_j), \quad \forall j \in J \tag{9}$$

$$\sum_{j \in J} \mathcal{H}_{jk} = 1, \quad \forall k \in K \tag{10}$$

$$s_{j'} \geq s_j + T(\mathcal{H}_{j,k}, d_j) - M(1 - \mathcal{W}_{jj'}) + \mathcal{Z}_{jj'} M, \tag{11}$$

$$\forall j \neq j', \ j, j' \in J$$

$$s_j \geq s_{j'} + T(\mathcal{H}_{j',k}, d_{j'}) - M(1 - \mathcal{W}_{jj'}) + \mathcal{Z}_{jj'} M, \tag{12}$$

$$\forall j \neq j', \ j, j' \in J$$

$$\mathcal{Z}_{jj'} + \mathcal{Z}_{j'j} = \mathcal{W}_{jj'}, \quad \forall j \neq j', \ j, j' \in J \tag{13}$$

$$\mathcal{W}_{jj'} \leq \mathcal{X}_{ij}, \ \mathcal{W}_{jj'} \leq \mathcal{X}_{ij'}, \forall j \neq j', \ \forall i \tag{14}$$

$$\mathcal{W}_{jj'} \geq \mathcal{X}_{ij} + \mathcal{X}_{ij'} - 1, \forall j \neq j', \ \forall i \tag{15}$$

$$\mathcal{X}_{ij}, \mathcal{H}_{jk} \in \{0, 1\}, \quad \forall i, \ \forall j \in J, \forall k \in K \tag{16}$$

$$\mathcal{Z}_{jj'}, \mathcal{W}_{jj'} \in \{0, 1\}, \quad \forall j \neq j', \ j, j' \in J \tag{17}$$

$$s_j \geq 0, \forall j \in J; \quad 1 \leq i \leq G \tag{18}$$

In this formulation, we input the number of hardware devices $G$ and LoRA configurations $K$. The rest are variables that the optimization instances solve. $J$ represents the set of jobs, $\mathcal{X}_{ij}$ is a binary variable that equals 1 if job $j$ is assigned to device $i$, $\mathcal{H}_{jk}$ is a binary parameter that indicates whether LoRA configuration $k$ belongs to job $j$. $s_j$ is the start time of job $j$, $\mathcal{Z}_{jj'}$ is a binary variable that ensures job ordering where $\mathcal{Z}_{jj'} = 1$ if job $j$ precedes job $j'$ and does not overlap in time, $\mathcal{W}_{jj'}$ is a binary variable indicating that whether job $j$ and $j'$ share at least one device. We employ $M$ as an auxiliary large constant in the ordering constraints (Ghirardi and Potts, 2005). During optimization for minimal makespan, the optimization instance would output the assignment of LoRA configurations to jobs ($\mathcal{H}$), the assignment of jobs to devices ($\mathcal{X}$), and the scheduling of jobs ($\mathcal{Z}$). $T()$ is the cost model used to estimate the training time of LoRA fine-tuning jobs; it is not a variable, but a function of the packed LoRA configurations $\mathcal{H}_{j,k}$ and the parallelism degree $d_j$, where $j$ is a job. The notations used in this section are listed in Table 3.

In our optimization setup, Equation (10) ensures that each LoRA configuration belongs to exactly one fine-tuning job; Inequalities (11), (12), and Equation (13) ensure that jobs sharing any devices do not overlap in time (Pinedo, 2008); Inequalities (14) and (15) help enforce the prior constraint by setting $\mathcal{W}_{jj'}$ to 1 when two jobs share at least one device (Pinedo, 2008). The makespan is represented as the latest job completion timestamp in Inequality (9). Equations (16), (17), and (18) define the scope of the variables. This optimization problem also has constraints on GPU

*Table 3.* Notation used in cost model formulation

| Symbol | Description |
|---|---|
| **Model-related parameters** | |
| $C$ | Memory load factor |
| $M_{\text{base}}$ | Memory required for the base model |
| $M_{\text{lora},k}$ | Memory required for LoRA configuration $k$ |
| $M_{\text{gpu}}$ | Total memory capacity of a GPU |
| $G$ | Number of available GPU devices |
| $K$ | Set of LoRA configurations |
| $J$ | Set of training jobs |
| $M$ | A large constant for scheduling ordering |
| $T()$ | Duration of job, estimated with cost model |
| **Optimization variables** | |
| $\mathcal{X}_{ij}$ | Binary variable: 1 if job $j$ runs on device $i$ |
| $\mathcal{H}_{jk}$ | Binary variable for LoRA assignment |
| $s_j$ | Start time of job $j$ |
| $r_k$ | LoRA rank of LoRA configuration $k$ |
| $\mathcal{Z}_{jj'}$ | Binary variable for scheduling order |
| $\mathcal{W}_{jj'}$ | Binary variable for device sharing |
| $d_j$ | Number of GPUs used by job $j$ |

memory usage per LoRA fine-tuning job and the number of GPUs that can be allocated across all concurrent jobs.

This optimization problem is NP-complete as it can be viewed as a variant of a 0-1 knapsack problem (Karp, 2009). In addition, our optimization formulation adds a new layer of complexity, as each job must first decide which LoRA configurations ($\mathcal{H}_{jk}$) to train and determine the associated degree of parallelism.

## A.1. Notation Table

Below is the notation table used in formulating the optimization problem:

## B. LoRA Memory Constraints

To ensure that the training job does not exhaust the available GPU memory, we impose a constraint on the total memory used by all LoRA configurations in a job, including the base model weights. The total memory usage must not exceed the GPU memory:

$$M_{\text{base}} + \sum_{k=1}^{K} M_{\text{lora},k} \leq c_{load} \times M_{\text{gpu}}$$

$M_{\text{base}}$ represents the memory cost of the base model, while $M_{\text{lora},k}$ represents the memory cost of a LoRA adapter on a device. As in other work on managing GPU memory (Kwon et al., 2023), the user can set a load factor $c_{load}$ to account for internal GPU fragmentation and to adjust GPU memory usage.

For each LoRA configuration $k$, the memory usage, $M_{\text{lora},k}$, is represented by an indicator variable to determine if the

user applies LoRA to those matrices. In each attention block, the user can apply LoRA to $Q$, $K$, $V$, and the output projection matrix; In each MLP block, the user can apply LoRA to the up, down, and gate projection matrix. We, therefore, write the LoRA memory usage as a sum of these 7 components:

$$M_{\text{lora},k} = \sum_{i=1}^{7} \mathbb{I}_i M_{\text{lora},i} \qquad (19)$$

Each index $i$ corresponds to one of the seven components that the user can apply LoRA to. For each component, the LoRA memory usage includes the memory required to store LoRA parameters, gradients, and activations:

$$M_{\text{lora},k} = M_{\text{lora\_param},k} + M_{\text{lora\_grad},k} + M_{\text{lora\_act},k} \qquad (20)$$

The memory for LoRA parameters, $M_{\text{lora\_param},k}$, is given by:

$$M_{\text{lora\_param},k} = n_{\text{layers}}(h_{\text{in}} \times r_{\text{lora},k} + h_{\text{out}} \times r_{\text{lora},k}) \times c_{\text{prec}}$$

In practice, this may change depending on memory-saving strategies such as activation checkpointing. Here, $n_{\text{layers}}$ is the number of layers, $h_{\text{in}}$ and $h_{out}$ represent the input and output dimensions of the projection matrix, which can take different values based on the model architecture and vary between attention and MLP blocks. $c_{\text{prec}}$ is the training precision, representing bytes per parameter.

The memory required for the gradients, $M_{\text{lora\_grad},k}$, is calculated as:

$$M_{\text{lora\_grad},k} = c_{grad} \times M_{\text{lora\_param},k} \times c_{\text{prec}}$$

$c_{grad}$ represents the scaling factor for storing gradient-related parameters. For example, this factor is set to 3 in the popular AdamW optimizer, representing momentum, velocity, and primary gradients.

Finally, the LoRA activation memory for each block, $M_{\text{lora\_act},i}$, is given by:

$$M_{\text{lora\_act},k} = b \times s \times r_{\text{lora},k} \times c_{\text{prec}}$$

Here $b$ is the batch size, and $s$ is the sequence length. This term represents the memory used to store intermediate activations during training. In LLM fine-tuning, the sequence length varies based on the workload. The standard practice is to set a maximum training length and split the training document if any data samples are too long, to prevent memory overflow. When computing memory consumption, we take the same approach and set the sequence length to the maximum length of the training samples.

Similarly, the activation memory for the base model can be computed by summing the activation of the embedding layer, the attention operator, and the feed-forward network in each layer. Depending on the implementation, other activations, such as those produced by layer norm, may also be computed and stored; our model can be easily adapted to those implementations. More details on computing the memory consumption for each of these four modules can be found in the Appendix:

$$M_{\text{base\_act}} = M_{\text{base\_emb}} + M_{\text{base\_attn}} + M_{\text{base\_mlp}}$$

The total memory for the base model is then calculated as:

$$M_{\text{base}} = M_{\text{base\_weights}} + M_{\text{base\_act}}$$

The computation in this section assumes a single-device setting. The section discusses how the parallelization strategy affects our memory constraints.

### B.0.1. PARALLELIZATION STRATEGY AND GPU CONSTRAINTS:

The prior section of constraints assumes that a full copy of the model is stored on each device. In practice, tensor parallelism, pipeline parallelism, and fully sharded data parallelism (FSDP) are popular strategies for parallelizing LLM training and are necessary for modern LLMs. The following section explains how we incorporate different parallelization strategies into our cost model. To accommodate tensor parallelism and pipeline parallelism, we can rewrite the memory cost associated with the LoRA adapters $i$ as follows:

$$M_{\text{lora\_param},k} = \frac{M_{\text{lora\_param},k}}{d_{\text{tp}} * d_{\text{pp}}}$$

The calculation is similar to that of the base model parameters and intermediate outputs. For FSDP, the model computes the following for different ZeRO optimizer levels. For ZeRO-1, the LoRA memory includes both the unsharded gradient and parameter memory and the sharded optimizer state:

$$M_{\text{lora},k}^{(1)} = M_{\text{lora\_grad},k}^{(1)} + M_{\text{lora\_param},k} + \frac{M_{\text{opt},k}}{d_{\text{fsdp}}}$$

For ZeRO-2, the gradient memory also includes the gradient term:

$$M_{\text{lora, k}^{(2)}} = M_{\text{lora\_param},k} + \frac{M_{\text{lora\_grad},k} + M_{\text{opt},k}}{d_{\text{fsdp}}}$$

For ZeRO-3, the memory includes all fully sharded components:

$$M_{\text{lora},k}^{(3)} = \frac{M_{\text{lora\_param},k} + M_{\text{lora\_grad},k} + M_{\text{opt},k}}{d_{\text{fsdp}}}$$

Then, the model will determine which concurrent training jobs to launch and assign each job a parallelization strategy, ensuring that the total GPU usage does not exceed the GPU constraints.

We apply this formulation to every memory cost computation and add a total GPU constraint where the model will solve for the number of jobs to launch concurrently, as well as the tensor parallel degrees and LoRA adapter configurations to be packed on each job:

$$\sum_j d_j \leq G$$

## C. Optimizing Packed LoRA Computation

### C.1. Inefficient Computation in Existing Frameworks

LLM pretraining frameworks, such as Megatron-LM (Shoeybi et al., 2019) and PyTorch (Zhao et al., 2023), and LoRA fine-tuning frameworks, such as PEFT (Mangrulkar et al., 2022) and Unsloth (Han et al., 2023), only support fine-tuning one configuration on a set of hardware at a time. Since LoRA adapters, when packed, share the same base model weights but have different adapter weights and inputs[1], a naive approach to support fine-tuning with packed LoRA adapters is to batch the computation of the base model and sequentially compute each LoRA adapter (Figure 2). However, this approach results in poor fine-tuning throughput due to low hardware utilization per LoRA adapter.

We profiled the fine-tuning performance using the naive approach as described above. We use Qwen-2.5-7B as the base model and apply a single LoRA adapter with batch size 1 on an A100 GPU as the baseline. The iteration time increases by $10\%$ as the batch size increases from 1 to 8. However, when we pack eight adapters into a fine-tuning job, and each adapter has a batch size of 1, the naive approach increases iteration time by $3.6\times$ compared to single LoRA tuning due to low hardware utilization in LoRA adapter computations. We also observe similar performance when fine-tuning Qwen-2.5-14B with two A100 GPUs and Qwen-2.5-32B with four A100 GPUs using TP. This confirms our hypothesis that the performance bottleneck is the sequential computation of LoRA adapters rather than the batched computation in the base model.

### C.2. Packed LoRA Kernels

We devise custom CUDA kernels for PLoRA to efficiently batch the computation of LoRA adapters in both forward and backward propagations. We carefully tile the LoRA

---

[1]We replicate the input tokens for each LoRA adapter at the beginning of each fine-tuning iteration.

matrices and group gradient computations across multiple adapters to improve hardware utilization and handle load balancing for heterogeneous LoRA adapters.

Given a set of LoRA adapters, we concatenate the LoRA adapters into a tensor and design kernels that can compute forward and backward passes for all LoRA adapters. Our key insight in designing performant kernels is to tile the concatenated tensor along the sequence or hidden dimensions if possible. The sequence dimension consists of the input token sequence multiplied by the batch size. We avoid tiling along the LoRA rank dimension because the rank can be as small as 8, and sharding on the smaller dimension prevents GPUs from fully utilizing their compute resources.

If we denote the dimension of LoRA A by $(d, r)$ and the dimension of LoRA B by $(r, k)$ (Figure 1), we typically have $d >> r$ and $r << k$. Previous work on serving multiple LoRA adapters (Chen et al., 2024) introduces a kernel that handles these two cases separately, by splitting the input dimension $d$ for LoRA A and the output dimension $k$ for LoRA B. While this is possible for the forward pass, backward propagation cannot simply reuse this strategy. When computing the activation gradients with respect to LoRA inputs, avoiding tiling over the rank dimension would require splitting the inner dimension for tiling. This strategy would require extra overhead to create a scratch buffer for each tile, additional indices for bookkeeping, and extra synchronization and reduction steps to accumulate intermediate results, undermining the benefits of tiling over large dimensions.

In our work, we tackle the challenge of efficient backward propagation implementation and use the following strategy to obtain performant kernels.

**CUDA kernel design.** We built upon CUTLASS for our packed LoRA kernels. Four cases should be considered separately for the upstream weights and input gradients of LoRA A and LoRA B. Below, we outline our backpropagation partitioning strategy in detail.

• Case 1: We compute the gradient for the weight of the LoRA B projection. We partition along the output dimension (k) of the LoRA B projection matrix to ensure that gradient computation correctly associates the $i$th LoRA slice with the corresponding input and output matrix slices. Tiling is done along the output dimension, and LoRA ranks $r_1$ and $r_2$ remain in each tile.

• Case 2: We compute the gradient for the input of the LoRA B projection. We tile over the sequence and LoRA rank dimensions of the upstream gradient, and reduce over the input hidden dimension.

• Case 3: We compute the gradient for the weight of the LoRA A projection. We tile over the sequence dimension of input activations and the output dimension of the upstream

gradients, using LoRA rank as the reduction axis.

• Case 4: We compute the gradient for the input of the LoRA A projection. We tile along the upstream gradients' sequence and LoRA rank dimensions and use the concatenated LoRA rank dimension for reduction.

**Kernel performance tuning.** To achieve high kernel performance across different hardware setups, we tune the ThreadblockShape, WarpShape, and InstructionShape parameters in CUTLASS to optimize performance (Thakkar et al., 2023; Gale et al., 2020). While optimal settings vary with both the underlying hardware architecture and the GEMM problem dimensions, we simulate workloads using model dimensions from widely used 3B and 7B models, along with sequence lengths ranging from 512 to 2048. We set the InstructionShape to $(16, 8, 16)$ to match the tensor core instruction shape on Ampere GPUs. For WarpShape, we empirically found that $(64, 64, 32)$ yields the best throughput on the A100, while $(16, 64, 32)$ performs best on the A10 without triggering memory errors. Based on these warp shapes, we configure ThreadblockShape as $(128, 128, 32)$ on the A100 and $(64, 64, 32)$ on A10 to ensure compatibility with WarpShape.

## D. Proof of greedy scheduling tail effect

**Algorithm analysis.** Algorithm 2 performs optimally in a streaming setting with an unlimited number of jobs, as it consistently selects the job with the highest concurrent LoRA fine-tuning throughput. However, in our setting with a finite number of jobs, this approach can lead to a tail effect: the final jobs may not fully utilize all available hardware resources, resulting in suboptimal overall throughput compared to the optimal solution. We now bound this tail effect.

**Theorem D.1** (Bounded Tail Effect with Algo 2). *Let $J$ be a set of jobs scheduled on $G$ GPUs. Let $j \in J$ be the last job that uses $D$ GPUs. Let $T_{last}$ be the fine-tuning time of the last job and $F$ be the makespan based on the job planner's schedule. Then, the approximation ratio ($AR$) of the job planner for the makespan optimization problem is upper bounded by: $AR \leq \frac{F}{F - T_{last} \cdot \frac{G-D}{G}}$.*

See Appendix D for the detailed proof. In practice, using experiment settings from §4, we find that PLoRA produces schedules with AR between 1.05 and 1.14.

*Proof.* We start by noting that before starting the last job, all $G$ GPUs are fully utilized by the definition of our greedy scheduling algorithm. Moreover, our algorithm offers a monotonicity condition: if a job using $x$ GPUs is scheduled, the next job in the optimal ordering requires no more than $x$ GPUs. This condition guarantees that no bubbles occur between jobs in fully loaded batches; the only underutilization

occurs in the final batch.

Define the total GPU work as $W = \sum_{j \in J} x_j t_j$. Recall that $t_{last}$ denotes the processing time of the last job and the cumulative time of the fully utilized jobs before starting the last job, be $F_{prev}$. Let $F = F_{prev} + t_{last}$ be the makespan of the job planner's schedule, and OPT be the makespan of an optimal schedule with full GPU utilization throughout. Then we can write:

$$W = F_{prev} \cdot G + t_{last}(G - D),$$

and the total makespan of the greedy schedule is

$$F = F_{prev} + t_{last}.$$

An optimal schedule (with full GPU utilization in every batch) must satisfy the following:

$$\text{OPT} \geq \frac{W}{G} = F_{prev} + t_{last} \frac{G - D}{G}.$$

Thus, we can bound the extra time incurred due to the bubble in the last batch by

$$F - \text{OPT} \leq [F_{prev} + t_{last}] - \left[ F_{prev} + t_{last} \frac{G - D}{G} \right] \tag{21}$$

$$= t_{last} \left( 1 - \frac{G - D}{G} \right) \tag{22}$$

$$= t_{last} \frac{D}{G} \tag{23}$$

This result quantifies the tail effect under asynchronous scheduling: the extra time is proportional to the fraction of idle GPUs for the last job. And we can obtain our final bound:

$$\frac{F}{\text{OPT}} \leq 1 + \frac{t_{last} \cdot \frac{D}{G}}{F_{prev} + t_{last} \cdot \frac{G-D}{G}} \tag{24}$$

which can be simplified to

$$\frac{F}{\text{OPT}} \leq \frac{F}{F - T_{last} \cdot \frac{G-D}{G}}.$$

$\square$

## E. Additional experiment results

### E.1. Sensitivity Analysis of LoRA configurations

We perform an extensive empirical study of the impact of hyperparameters on model quality with LoRA fine-tuning. The detailed experimental setup is described in §4. We use

*Table 4.* This table shows the optimal hyperparameter configuration we found during the hyperparameter sweep.

| | 3B | | | | 7B | | | |
|---|---|---|---|---|---|---|---|---|
| Task | Rank | LR | BS | $\alpha$ | Rank | LR | BS | $\alpha$ |
| mrpc | 16 | 4e-5 | 1 | 1 | 32 | 6e-5 | 1 | 1 |
| cola | 64 | 4e-4 | 1 | 0.25 | 32 | 8e-5 | 1 | 0.5 |
| wnli | 32 | 2e-4 | 2 | 1 | 32 | 2e-4 | 4 | 0.5 |
| gsm8k | 32 | 1e-4 | 2 | 1 | 16 | 3e-4 | 1 | 1 |

*Table 5.* This table analyzes LoRA hyperparameter sensitivity. The base model is QWen-2.5-7B. We only tune one hyperparameter and keep the others fixed. The results are the maximum accuracy differences by tuning the chosen hyperparameter.

| Task | LR | BS | Rank | $\alpha$ |
|---|---|---|---|---|
| mrpc | 8.5% | 10.0% | 6.4% | 4.9% |
| cola | 14.2% | 8.5% | 13.1% | 5.9% |
| wnli | 6.8% | 11.3% | 5.4% | 5.5% |
| gsm8k | 5.0% | 3.2% | 4.5% | 2.5% |

QWen-2.5 (Yang et al., 2024) as the base model and report the zero-shot accuracy on the following benchmarks in our experiments: **GSM8K** (Cobbe et al., 2021) for mathematical reasoning; **mrpc** (Wang et al., 2018) for language understanding; **cola** (Wang et al., 2018) for commonsense reasoning; and **wnli** (Wang et al., 2018) for logic reasoning.

**Observation #1: Hyperparameters strongly influence LoRA model quality.** We begin by varying only one hyperparameter at a time (Table 5) while fixing others to the optimal configuration in Table 4, we find accuracy differences of up to 14.2% (learning rate), 11.3% (batch size), 13.1% (LoRA rank), and 5.9% (LoRA $\alpha$). Second, by evaluating 120 LoRA configurations (Table 6) built from the search space in Table 1, we observe strong variance in model accuracy.

We also study how the best LoRA configurations vary across different LoRA fine-tuning workloads, which are evaluated using both QWen-2.5-3B and QWen-2.5-7B as base models. The best LoRA configurations for different workloads are listed in Table 4.

*Table 6.* Model quality of QWen-2.5-7B with the base model only, the worst, and the best LoRA hyperparameter configuration across various tasks. $\Delta$ denotes the improvement in accuracy from the best configuration to the base model.

| | Base | Worst | Best | $\Delta$ |
|---|---|---|---|---|
| mrpc | 64.1% | 57.1% | 70.0% | +5.9% |
| cola | 62.7% | 61.5% | 80.2% | +18.5% |
| wnli | 78.8% | 74.7% | 84.5% | +5.7% |
| gsm8k | 70.8% | 71.2% | 79.8% | +9.0% |

*Table 7.* The normalized throughput improvement of packed LoRA kernels over sequential LoRA computations. The first number in each cell represents the throughput speedup in the forward pass, while the second number represents that in the backward pass.

| Num. LoRA | 3B Attention d = 2048 | 3B MLP 11008 | 7B Attention 3584 | 7B MLP 18944 |
|---|---|---|---|---|
| 2 | 2.00x / 2.01x | 1.98x / 1.98x | 1.90x / 1.92x | 1.99x / 1.99x |
| 8 | 7.98x / 7.96x | 7.60x / 7.67x | 7.51x / 7.92x | 7.77x / 7.80x |
| 32 | 29.0x / 30.0x | 26.5x / 26.9x | 26.7x / 31.2x | 28.4x / 28.7x |

**Observation #2: Optimal LoRA configurations vary across tasks and base models.** Table 4 shows that the best hyperparameter settings for LoRA fine-tuning depend on both the downstream task and the base model. For instance, with QWen-2.5-3B, the best configuration for mrpc is [16, 4e-5, 1, 1], while gsm8k requires [32, 1e-4, 2, 1]; applying the mrpc configuration to gsm8k reduces accuracy by 7.4%. Similarly, transferring the best configuration for cola on QWen-2.5-7B to QWen-2.5-3B decreases accuracy by 3.6%. These results highlight that effective LoRA fine-tuning requires workload- and model-specific configurations.

**Observation #3: LoRA fine-tuning benefits from small batch sizes.** As shown in Table 4, LoRA consistently achieves higher accuracy with smaller batch sizes ($\leq 4$), a trend also reported in prior work (Zhao et al., 2024; Hu et al., 2021a; Fomenko et al., 2024). Smaller batches reduce gradient variance when only a fraction of parameters are updated, which improves convergence and generalization (Hu et al., 2021a).

### E.2. Microbenchmarks

#### E.2.1. PACKED LORA KERNEL PERFORMANCE.

We examine the performance of our customized LoRA kernels in various workloads on A100 GPUs. Consider a LoRA tensor with a shape $[r, d]$, where $r$ is the LoRA rank and $d$ is the hidden dimension in the base model. We vary both $r$ and $d$ to evaluate the computational efficiency of the packed LoRA kernel. We first fix $r = 64$, set $d$ to different values corresponding to the hidden dimensions of the Attention and MLP layers in QWen-2.5-3B and QWen-2.5-7B, and set the batch size to 1. We pack different numbers of LoRA computations into a kernel (ranging from 2 to 32) and compare the forward and backward computation performance with a sequential baseline.

Table 7 reports the throughput improvement normalized to the performance of the baseline. As we increase the number of packed LoRA computations from 2 to 32, our packed LoRA kernels exhibit close to linear speedups over the baseline in both forward and backward propagation. This trend holds across a wide range of hidden dimensions

*Table 8.* In this table, we show the throughput improvement of attention and MLP forward and backward LoRA kernels using sequence length 1024 on A10 GPUs as we scale up the number of concurrent LoRA adapters. The first number in each cell represents the throughput improvement (FLOP/s) of the forward pass, while the second number represents the backward pass.

| # LoRA
Dim | 3B Attention
2048 | 3B MLP
11008 | 7B Attention
3584 | 7B MLP
18944 |
|---|---|---|---|---|
| 2 | 1.98x/1.98x | 1.9x/1.86x | 1.94x/1.97x | 1.98x/1.90x |
| 8 | 7.65x/7.55x | 7.52x/7.42x | 7.48x/7.4x | 7.44x/7.5x |
| 32 | 25.95x/26.09x | 25.87x/26.14x | 27.24x/26.45x | 26.78x/26.97x |

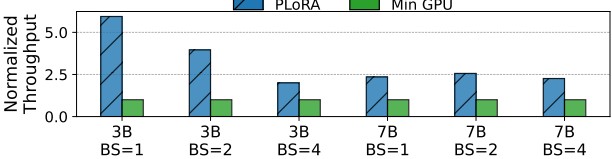

*Figure 7.* This figure shows the LoRA fine-tuning throughput for various models and batch sizes on A10 GPUs, normalized to the Min GPU baseline.

(2048-18944), and LoRA ranks (8-128).

In Table 8, we present kernel speedup as we scale up LoRA forward and backward kernels on A10 GPUs.

### E.3. Fine-tuning Throughput on A10 GPUs

We evaluate PLoRA on A10 GPUs using the QWen-2.5-3B and QWen-2.5-7B models, with a LoRA rank of 32. The throughput of LoRA fine-tuning jobs is shown in Figure 7, and the performance is normalized to the Min GPU baseline. PLoRA achieves $5.94\times$ speedup for 3B and $2.56\times$ speedup for 7B. The throughput improvement is lower than that on A100 GPUs, which is expected because A10 GPUs have less GPU memory capacity than A100 GPUs and, therefore, can pack fewer LoRA adapters in LoRA fine-tuning jobs.

We also evaluate PLoRA on base models with QLoRA (Dettmers et al., 2023), which quantizes the weights of the base model to 4 bits. QLoRA reduces the base model's GPU memory usage, freeing up more memory for LoRA adapters. We enable QLoRA in PLoRA and evaluate the performance with QWen-2.5-7B. We use LoRA with a rank of 32 and a batch size of 1 in all LoRA configurations. PLoRA achieves $4.72\times$ speedup compared to standard QLoRA fine-tuning with a single LoRA. This experiment shows that quantization, an orthogonal approach to boost LoRA fine-tuning efficiency, can work with PLoRA to further improve fine-tuning throughput by packing more LoRA adapters in LoRA fine-tuning jobs.

