# OpenReview forum: "PLoRA: Efficient Concurrent LoRA Training for Large Language Models"
_ICML.cc/2026/Conference — ICML 2026 regular_

### Official Review · Reviewer_66ay · 2026-03-05

**Soundness:** 3
**Presentation:** 4
**Significance:** 3
**Originality:** 3
**Overall Recommendation:** 4
**Confidence:** 2

**Summary:**

This paper propose PLoRA, which packs multiple LoRA adapters into a single fine-tuning run and trains them concurrently while sharing the same frozen base model. PLoRA combines offline packing/scheduling (formulated via knapsack/ILP-style optimization and solved with approximation methods) with an online execution engine for deployment, and introduces high-performance GPU kernels tailored for packed LoRA. Across multiple models and settings, PLoRA reports up to 12.8× higher training throughput and 7.52× lower overall fine-tuning makespan, while also enabling better configuration choices that can improve quality within a given search space.

**Compliance With Llm Reviewing Policy:**

Affirmed.

**Final Justification:**

I would like to thank the authors for their detailed rebuttal and for carefully addressing my concerns. Their response has resolved most of the issues raised in my initial review and has clarified several important points. Given these improvements and clarifications, I believe my current assessment remains appropriate.

**Key Questions For Authors:**

1. In addition to the hyperparameter search space considered in the paper, the number of training epochs/steps is also a crucial factor because it determines how long each run lasts. How does the method handle the case where different packed tasks/configurations require different numbers of epochs/steps? If some runs finish early, wouldn’t that reintroduce under-utilization? Conversely, if the finished runs continue training together with the pack, could this cause over-training and wasted computation?

**Limitations:**

The paper evaluates on a limited set of tasks, and the chosen context length (seq len = 1024) is relatively short, making it hard to assess whether the approach generalizes to broader scenarios and a wider range of tasks.

**Strengths And Weaknesses:**

Strength:
1. **Well-motivated problem with empirical evidence:** The paper provides profiling results showing clear under-utilization in LoRA training, grounding the motivation for concurrent LoRA training.
2. **End-to-end system design:** PLoRA presents a practical architecture spanning offline planning (packing, resource allocation, queue generation) and online execution (resource monitoring, job launch, checkpoint handling), which strengthens deployability.
3. **Principled optimization formulation with approximation guarantees:** This paper approximates makespan minimization via throughput maximization, and proposes decomposition-based algorithms (DTM / job planner).
4. **Strong and broad evaluation (throughput, makespan, and quality):** Experiments cover multiple model families and tasks, reporting large makespan reductions and throughput gains, and showing that PLoRA outperform common LoRA defaults (with sizable gains on some tasks).

Weakness:
1. The paper lacks a clear description of the cost model. Is it, for example, a learned predictor (e.g., a neural network) for estimating training cost? More generally, how does the system predict the runtime/memory cost for a given task and a set of LoRA hyperparameters?
2. Is there any analysis of how differences among the downstream tasks (and their corresponding LoRA adapters) affect concurrent training? If tasks are highly heterogeneous, could there be interference across tasks in a packed run? The experiments mainly include MRPC, CoLA, WNLI, and GSM8K; it would be helpful to evaluate on more domain-specific tasks with larger task/domain gaps to validate robustness under high heterogeneity.

---

> ### Author Rebuttal · Authors · 2026-03-31
>
> We thank the reviewer for their constructive and insightful feedback. Below, we address each weakness and question.
>
> W1: Cost model
>
> The cost model is an analytical model, not a learned predictor. It computes memory and runtime estimates from first principles:
>
> Memory estimation (detailed in Appendix B): For each LoRA configuration k, total memory is decomposed as:
> - Base model memory: M_base = M_base_weights + M_base_act (weights + activations)
> - LoRA memory per adapter: M_lora,k = M_lora_param,k + M_lora_grad,k + M_lora_act,k
>   - Parameters: computed from model dimensions (h_in, h_out), number of layers, rank, and precision
>   - Gradients: proportional to parameter count (scaled by optimizer factor, e.g., 3x for AdamW)
>   - Activations: computed from batch size, sequence length, and rank
>
> The model accounts for parallelization strategies (TP, PP, FSDP/ZeRO-1/2/3) by adjusting memory per device accordingly (Appendix B.1.1).
>
> Training time T(H_j,k, d_j) is calibrated from the first few steps of actual execution. Since LoRA fine-tuning has a stable per-iteration time, these early measurements reliably predict the full training duration.
>
>
> W2: No analysis of how task heterogeneity affects concurrent training.
>
> In PLoRA's packed fine-tuning, each adapter maintains its own independent parameters, optimizer state, training data, and loss function (Section 2.4, Figure 2). As long as total memory fits within GPU constraints (enforced by the planner), each adapter's computation is identical to running it in isolation, just more efficiently batched. Our four tasks, spanning mathematical reasoning and language understanding, have quite different data characteristics and loss landscapes.
>
> Q1: How does the method handle different packed tasks requiring different numbers of epochs/steps? If some runs finish early, wouldn't that reintroduce under-utilization? If finished runs continue, could this lead to overtraining?
>
>
> We thank the reviewer for the question. In the hyperparameter tuning literature, there is a standard way to handle this by breaking a long-running job into smaller jobs to reduce the bubbles arising from different training lengths [1]. This technique can also be applied to PLoRA, as checkpoints can be saved and reloaded in a future job.  In addition, PLoRA does not continue training completed adapters; each adapter tracks its own step count and stops when complete. The packed job continues only for the remaining active adapters.
>
> We also note that this constraint can also be solved at The Packing Planner level by: (1) grouping configurations with similar expected training durations into the same packed job (adding a duration-similarity constraint to the ILP), (2) using the Job Planner (Algorithm 2) to predict completion events and schedule the next batch of jobs to coincide with freed resources (Please refer to the response to reviewer DN1f’s Q1 for more results).
>
> [1]. A system for massively parallel hyperparameter tuning. Li, et. al. MLSys 2020

---

> > ### Author Rebuttal · Reviewer_66ay · 2026-04-02
> >
> > I would like to thank the authors for their detailed rebuttal and for carefully addressing my concerns. Their response has resolved most of the issues raised in my initial review and has clarified several important points. Given these improvements and clarifications, I believe my current assessment remains appropriate.

---

### Official Review · Reviewer_do8B · 2026-03-11

**Soundness:** 2
**Presentation:** 1
**Significance:** 2
**Originality:** 2
**Overall Recommendation:** 3
**Confidence:** 4

**Summary:**

This paper presents PLoRA, a system designed to improve the efficiency of multi-LoRA training. The authors conduct an empirical study revealing that LoRA fine-tuning favors small batch sizes, which leads to GPU underutilization, particularly when training multiple LoRA adapters. To solve it, PLoRA packs multiple heterogeneous LoRA configurations into a single fine-tuning job. The scheduling problem is formulated as a knapsack problem and solved with an approximation algorithm for offline planning. During online execution, PLoRA employs specialized GPU kernels for efficient multi-LoRA training. Experimental results demonstrate significant reductions in makespan and improvements in training throughput.

**Compliance With Llm Reviewing Policy:**

Affirmed.

**Final Justification:**

Thank you for the rebuttal. However, I will maintain my negative score since my concerns still remain.

---

Below replies the second rebuttal.

- Regarding the problem you consider: IMHO, a core challenge in *scheduling* is that fine-tuning requests may come and/or finish dynamically. However, the paper evaluates with a fixed number of LoRA adapters (and does not consider metrics for serving), which fails to distinguish your work from prior works. I am afraid heavy re-writing is necessary here.

- Regarding sequence length: As far as I concern, 1024 is not a standard sequence length for fine-tuning. For examle, domain-specific QA requires fine-tuning over domain-related documents. Besides, the sequence chunk length in LobRA is not the sequence length for training.

- Regarding hybrid parallelism and multi-node training: You can clarify that the scope of your paper does not include hybrid parallelism and multi-node training. However, please do not make claims like "complex hybrid parallelism is used only in large-scale pretraining runs and is not widely used in fine-tuning". For state-of-the-art models with a large number of parameters, hybrid parallelism and multi-node training are still necessary.

**Key Questions For Authors:**

- Q1: What are the specific novel contributions of PLoRA that differentiate it from existing multi-LoRA systems like mLoRA, LoRAFusion, and LobRA? It would be better to provide a comparison against these systems and stronger baselines that use optimally tuned hybrid parallelism strategies (combining DP, TP, and PP) on a common benchmark to substantiate claims of performance leadership.

- Q2: How does the planner handle dynamic workload scenarios with heterogeneous training length requirements? Does the current system assume a fixed number of configurations and the same training steps for all configurations inside a single packed job, and how to tackle the cases where some LoRA tasks finish earlier than others in the job?

- Q3: How does PLoRA's scheduling strategy scale with increasing numbers of LoRA configurations? Case studies and empirical results showing the scheduler's performance as the number of configurations and GPUs grow would make the contribution clearer.

- Q4: How about the scheduler’s solving time and solution quality under more complex scenarios, including hybrid parallelism strategies (DP+TP+PP), larger batch sizes, and larger clusters (e.g., 32+ GPUs)? What are the practical limits and scope of the current scheduler design under such conditions?

**Limitations:**

Yes

**Strengths And Weaknesses:**

### Strengths

- S1: Strong Empirical Foundation. Through a detailed empirical study (Section 2.3 and Appendix E), the authors convincingly demonstrate the tension between LoRA's preference for small batch sizes and the need for high arithmetic intensity to fully utilize modern GPUs. This analysis effectively motivates the design of PLoRA.

- S2: Practical and Effective Design. The proposed two-stage design (offline planner + online engine) is practical and addresses the identified performance bottlenecks head-on. The experiment results are impressive.

- S3:  Demonstrated Generalizability. The paper thoughtfully discusses PLoRA's applicability to other LoRA-variants like QLoRA and parallel strategies like FSDP, showing that the core ideas are transferable to other PEFT methods and common distributed training strategies, enhancing the work's impact.

### Weaknesses

- W1: Insufficient Discussion of Related Work and Novelty Concerns. The paper lacks a thorough discussion of its relationship to recent concurrent work on multi-LoRA training systems. Specifically:
    - LoRAFusion (Zhu et al., EuroSys'26) also targets the multi-LoRA scenario with new CUDA kernels and kernel fusion, and introduces a scheduler to mitigate load imbalance from variable sequence lengths.
    - LobRA (Lin et al., VLDB’25) tackles the challenge of heterogeneous sequence lengths across tasks by employing static heterogeneous parallel strategies and a dynamic workload-balanced data dispatcher.

    The paper should clarify how PLoRA's approach is different from or superior to these existing works. What is the specific contribution that sets it apart? A quantitative comparison against these systems on a common benchmark would be necessary to substantiate claims of novelty and performance leadership. Without this, it's difficult to assess the core contribution of PLoRA.

- W2: Limited Experimental Scale and Baselines. The experiments are limited to 8 GPUs and a fixed set of 120 LoRA configurations, without evaluating distributed training scenarios and hybrid parallel strategies for optimal configuration of baselines (e.g., only TP=8 for “MAX GPU” baseline instead of combining DP, TP, and PP). The workloads use a fixed sequence length of 1024, ignoring the significant heterogeneity of sequence lengths across different configurations, a key challenge highlighted by works like LobRA. The largest model tested is 32B, leaving scalability to larger models like 70B and multi-node training unconcerned.

- W3: Lack of Scalability Studies. While the paper demonstrates kernel performance scaling with up to 32 LoRA adapters in microbenchmarks, the end-to-end evaluation is limited to 120 fixed LoRA configurations. The paper does not validate how the scheduling strategy scales with increasing numbers of LoRA configurations. This is important for understanding the scheduler's applicability to different system workloads.

- W4: Scheduler Practicality Concerns. PLoRA's scheduler enumerates all possible parallelism strategies to solve the ILP problem (Section 3.2). While the paper claims this is practical for their evaluated scenarios, I have concerns about its practicality under larger and more complex workloads and configurations. The paper does not validate scheduler solving time under more complex scenarios: large batch sizes, distributed multi-GPU settings, and hybrid parallelism strategies (e.g., combining DP, TP, and PP). While Section 4.4 shows that LoRA fine-tuning tends to prefer small batch sizes, from the perspective of efficient system design, the paper should provide empirical measurements of scheduler overhead in more diverse scenarios to show the practicality scope and limitations of the current scheduler design.

- W5: Unclear Writing and Experimental Focus. Several aspects of the paper's presentation could be improved: (1)  While kernel optimization contributes significantly to speedup (up to 30× in microbenchmarks), the main text focuses primarily on scheduling design. The paper lacks case studies that clearly illustrate how scheduling decisions contribute to performance improvements. (2) The hyperparameter sensitivity experiment (Section 4.4) evaluates model quality across 120 fixed configurations using PLoRA, but does not compare against baseline approaches or demonstrate how this validates the system design. Given that the core contribution of PLoRA is accelerating multi-LoRA training by system design instead of algorithm innovation of hyperparameter searching, the connection between this experiment and the paper's core contributions is unclear. (3) The paper discusses fine-tuning-as-a-service scenarios with dynamic workloads (Section 1), but the evaluation only considers 120 fixed LoRA configurations. The paper does not explain how PLoRA handles runtime workload changes or whether the scheduler can adapt to dynamic workload.

- W6: Limited Kernel Portability. The customized CUDA kernels are primarily tuned for the Ampere architecture (A100, A10) with specific CUTLASS parameters (ThreadblockShape, WarpShape, InstructionShape). The paper does not demonstrate whether these kernels generalize to other GPU architectures (e.g., Hopper H100) or how much re-tuning would be required. Given that optimal settings vary with both hardware architecture and GEMM problem dimensions, this limits the method's portability.

---

> ### Author Rebuttal · Authors · 2026-03-31
>
> We thank the reviewer for their constructive and insightful feedback. Below, we address each weakness and question.
>
> W1 & Q1: Related works.
>
> We thank the reviewer for pointing out these concurrent works; we will include them in the revised paper. We will discuss how PLoRA addresses a fundamentally different and complementary problem space compared to these systems:
>
> mLoRA uses pipeline parallelism to scale LoRA fine-tuning across devices. LoRAFusion uses a similar setup but focuses on kernel-level fusion to reduce redundant memory accesses, and on grouping adapters and batches of samples across jobs to reduce pipeline bubbles and load imbalance. LobRA targets a related but distinct problem: efficiently handling heterogeneous sequence lengths across multiple fine-tuning tasks.
>
> PLoRA, in contrast, jointly optimizes *which configurations to pack together* and *how many GPUs to allocate per job*, a combinatorial optimization that LoRAFusion's adapter grouping heuristic does not address.
>
> These systems are largely complementary: mLoRA and LoRAFusion's kernel fusion could be applied within PLoRA's packed execution only if a user chooses pipeline parallelism (Please refer to the response to Reviewer Nteo’s W2 and Q1 for more details), and LobRA's data dispatching could handle sequence heterogeneity within PLoRA's fixed-configuration packs. But neither can schedule and decide which configurations to pack together, and their corresponding parallelism strategy. These points capture the novelty and the unique contribution of PLoRA.
>
> W2: Limited Experiment scale
>
> We acknowledge these limitations and address each point:
>
> Parallelism strategies: We build PLoRA on top of a state-of-the-art open-source fine-tuning platform, torchtune[2], and natively support both TP and FSDP, the default options in popular training platforms [1][2][3]. In practice, complex hybrid parallelism is used only in large-scale pretraining runs and is not widely used in popular LoRA fine-tuning libraries [1][2][3].
>
> Sequence length: We would like to clarify that the sequence length here refers to the maximum sequence length and is therefore heterogeneous across different workloads. The exact sequence length depends on the dataset / workload being run.
>
> Model scale: We cover a wide range of model sizes from 3B to 32B, given the resources we have.
>
> [1]. Axolotl: Open Source LLM Post-Training. https://github.com/axolotl-ai-cloud/axolotl
>
> [2]. torchtune: PyTorch's finetuning library. https://github.com/meta-pytorch/torchtune
>
> [3]. Unsloth, Han et al. https://github.com/unslothai/unsloth
>
> W3 & Q3 & W4 & Q4: Experiment Scalability
>
> We conduct the following experiment to show that the scheduling overhead would remain negligible:
>
> For a 1000+ configuration sweep on a 32-GPU cluster, our planner's overhead remains negligible. Modern ILP solvers like Gurobi are designed to handle optimization problems with 10^7 or more variables.
> To provide concrete evidence, we measured the time our ILP solver (Gurobi) takes to solve a single packing instance (Line 6) as we increase the number of LoRA configurations.
> ILP instance solver time:
> | Number of LoRA Adapters | Gurobi Solver Time / Iteration |
> |-------------------------|--------------------------------|
> | 120                     | 0.214s                         |
> | 1200                    | 0.215s                         |
> | 12000                   | 0.215s                         |
>
> As we see, the solver can handle 12K LoRA configurations in the same time it takes to handle 120 configurations; therefore, the total time the solver takes to decompose 12K LoRA configurations into jobs is less than 4 minutes.
>
> W4 & Q4: Batch sizes.
>
> Batch size: We conduct a comprehensive empirical study to identify the best batch sizes (Section 2). As we demonstrate in Section 2.3 and as supported by prior works [1][2][3], LoRA fine-tuning requires small batch sizes to achieve optimal model quality.
>
> [1]. LoRA Land: 310 Fine-tuned LLMs that Rival GPT-4, Zhao et. Al. https://arxiv.org/abs/2405.00732
>
> [2]. A Note on LoRA. Fomenko et. al. https://arxiv.org/abs/2404.05086
>
> [3]. LoRA Without Regret. Schulman et. al. https://thinkingmachines.ai/blog/lora/
>
> W5: More experiments
>
> We note that we already go beyond prior work (e.g., LoRAFusion) by creating a practical, heterogeneous workload to demonstrate its applicability via a thorough workload and hyperparameter sensitivity study (Section 2). **Due to space constraints, we kindly refer the reader to the responses to Reviewer DN1f’s W1 and Q1, and to Reviewer Nteo’s W2 and Q1, for more experiment details.**
>
> W6: Kernel portability
>
> Yes, kernels are, by design, hardware-specific. However, the principles behind kernel writing generalize across hardware.

---

> > ### Author Rebuttal · Reviewer_do8B · 2026-04-03
> >
> > - W1:
> >
> > GPU allocation is also considered in LobRA, and both LobRA and LoRAFusion consider data batching. It is for sure that you have different problem formulations, however, it is important to clarify how they are fundamentally different and why your approach is superior.
> >
> > - W2:
> >
> > > We build PLoRA on top of a state-of-the-art open-source fine-tuning platform, torchtune[2], and natively support both TP and FSDP, the default options in popular training platforms [1][2][3].
> >
> > While you acknowledge that torchtune supports FSDP and TP, the "Max GPU" baseline is restricted to TP=8. Besides, platformas like Unsloth provide multiple backends, and torchtune is not the default/best option as far as I concern (e.g., see this [link](https://unsloth.ai/docs/basics/faster-moe#automatic-backend-selection)).
> >
> > Moreover, my concern about limited baselines is not addressed. The "Min GPU" and "Max GPU" baselines are overly simple given the prior works.
> >
> >
> > > the sequence length here refers to the maximum sequence length and is therefore heterogeneous across different workloads. The exact sequence length depends on the dataset / workload being run.
> >
> > The sequence length is set to 1024 in the evaluation, which indicates that the sequences for all datasets are truncated to be at most 1024. I'm afraid this is too short to represent real-world workloads.
> >
> > - W3 & W4:
> >
> > The solver time is reported as per iteration. What does an iteration indicate? In addition, I'm confused by a discrepancy that the total time for 12K LoRA configurations is less than 4 minutes in the rebuttal while the time for 120 configurations is 10 minutes in the paper.
> >
> > If the overhead for larger batch sizes, distributed multi-GPU settings, and hybrid parallelism strategies cannot be assessed, then I suggest making the scope clearer in the paper.
> >
> > - W5:
> >
> > > We note that we already go beyond prior work (e.g., LoRAFusion) by creating a practical, heterogeneous workload to demonstrate its applicability via a thorough workload and hyperparameter sensitivity study (Section 2).
> >
> > This statement is confusing. How Section 2 makes your work go beyond prior works?
> >
> > Moreover, regarding W5, I concern that paper's focus (e.g., whether kernel optimization is the major contribution, and whether Section 1 should be refrained to static workloads) and suggest refining the writing, not asking for more results.

---

> > > ### Author Response · Authors · 2026-04-07
> > >
> > > We thank the reviewer for the detailed feedback.
> > >
> > > W1: To the best of our knowledge, we are the first to answer the question: **Given a set of LoRA adapters, how do you pack and schedule them?**
> > >
> > > **To answer the above question, we are the first to investigate packing and scheduling in multi-LoRA training, enabling a new use case for concurrent LoRA training.** Therefore, no head-to-head comparisons are readily available. For instance, existing works (LobRA and LoRAFusion) generally fix the number of LoRA adapters. This can lead not only to underutilization but also to OOM in certain cases. We are the first to explicitly optimize this packing and scheduling decision, in conjunction with lower-level optimizations to support fast, concurrent training.
> > >
> > > In comparison, concurrent works such as mLoRA and LoRAFusion target efficient kernels for pipeline parallelism but operate with a fixed number of adapters; PLoRA operates at one level above as an oracle scheduler. LobRA optimizes deployment and data dispatch for a given batch of FT tasks under sequence-length heterogeneity.
> > >
> > > If a user wants to train a set of LoRA adapters, PLoRA is the only option for breaking it into smaller, viable jobs. Once they have these jobs, they can then consider executing with the kernels provided by LoRAFusion or mLoRA if and only if they use pipeline parallelism (which is not widely supported in modern LoRA training libraries, such as Axolotl, Torchtune, and Unsloth). Otherwise, they can still opt for PLoRA’s execution engine for widely supported TP and FSDP scenarios.
> > >
> > > W2:  **We included a pointer to our rebuttal to another reviewer with new baseline results due to space constraints. We copy and paste them below:**
> > >
> > > To demonstrate the gains unlocked by the new adaptive scheduling layer introduced in our work, we compare the PLoRA scheduler with LoRAFusion kernels to a simple greedy scheduler that packs the default 4-adapter setup if possible, and halves it if OOM. Note that in order to accommodate LoRAFusion’s limitation on only supporting pipeline parallelism, we further restrict our evaluation to PP only. We follow a similar setup and evaluate Llama-3.1-8B and Qwen-2.5-32B across our LoRA config mix. We report the speedup in makespan in the following table
> > >
> > > | Makespan Reduction (↑)     | Llama-3.1-8B | Qwen-2.5-32B |
> > > |----------------------------|--------------|--------------|
> > > | LoRAFusion + PLoRA         | 2.62x        | 2.29x        |
> > > | LoRAFusion + Greedy        | 2.31x        | 2.05x        |
> > > | Min GPU                    | 1.00x        | 1.00x        |
> > >
> > > Sequence length: 1024 is a standard sequence length for non-reasoning models. For instance, mLoRA uses a sequence length of 512, while LobRA starts considering sequence chunks of length 256. In addition, frameworks such as Axolotl (https://docs.axolotl.ai/docs/config-reference.html) and TRL (https://huggingface.co/docs/trl/sft_trainer) set default lengths of 512 and 1024, respectively. Therefore, we believe this sequence length is appropriate. We will include an ablation study on how longer sequence length affect makespan in the final draft.
> > >
> > > W3&W4: Here, each iteration refers to line 8 in Algorithm 1, and we measure only the solver time. The end-to-end latency includes a one-time amortized cost for profiling representative jobs in the configuration space before we run the solver.
> > >
> > > Thus, our ablation study shows that if someone wants to scale up the number of configurations from 120 to 12k (e.g., by using a larger batch size or exploring different parallelism strategies), the scheduler can handle it with ease.
> > >
> > > Regarding the scope of our paper, we would like to note two points. First, we note that **our configuration space is already the largest-scale** compared to existing works, and carefully constructed, backed by design space analysis (Section 2 and Appendix E). We do not believe most users would need to explore a larger space. Therefore, the scheduling experiments only tackle a hypothetical scenario.
> > > Second, as we already pointed out in our response: **In practice, complex hybrid parallelism is used only in large-scale pretraining runs and is not widely used in popular LoRA fine-tuning libraries (such as Axolotl, Torchtune, and Unsloth).**
> > >
> > > That said, we are happy to clarify that the scope of our paper does not include complex hybrid parallelism and multi-node training to address the reviewer’s concern about writing.
> > >
> > > W5: In Section 2, we are the first to systematically study “What is a good design space for LoRA workload” by analyzing how hyperparameters affect LoRA performance. In comparison, concurrent works use fixed LoRA configurations (including arbitrarily determined batchsize and parallelism strategy).
> > >
> > > We also thank the reviewer for the writing feedback and will edit the draft to reflect our contribution.

---

### Official Review · Reviewer_Nteo · 2026-03-12

**Soundness:** 3
**Presentation:** 3
**Significance:** 3
**Originality:** 3
**Overall Recommendation:** 4
**Confidence:** 4

**Summary:**

The authors try to study system efficiency for LoRA fine-tuning when many adapters need to be trained. They claim that current LoRA training pipelines are highly inefficient because each adapter is trained independently, even though single LoRA jobs underutilize GPU compute and memory. To solve this problem, the authors introduce PLoRA, a system for concurrent LoRA fine-tuning. It packs multiple LoRA configurations into shared jobs and uses an offline planner to optimize packing and hardware allocation. Finally, it utilizes custom packed LoRA kernels to improve execution efficiency. Experiments on Qwen2.5 and LLaMA model families report large reductions in end-to-end makespan and large job-level throughput gains, while also showing that searching more LoRA configurations can improve downstream task quality relative to default LoRA settings.

**Compliance With Llm Reviewing Policy:**

Affirmed.

**Final Justification:**

The authors have responded adequately to my questions.

**Key Questions For Authors:**

1. Can you compare PLoRA against stronger systems baselines beyond Min GPU and Max GPU, for example simpler concurrent packing heuristics or alternative scheduling methods without the full planner?
2. How robust is the cost model used by the planner across different tasks, ranks, batch sizes, and model families?
3. Can you clarify more explicitly that the model-quality gains come from being able to search more configurations within a fixed budget, rather than from any direct algorithmic quality improvement in a single LoRA run?

**Limitations:**

yes

**Strengths And Weaknesses:**

**Strengths**
- There has been a lot of work on concurrent LoRA serving, but much less on efficiently training many LoRA adapters. That makes the problem setting relevant and timely.
- Packing multiple LoRA adapters into a shared run is well motivated by the observation that single LoRA jobs have low SM occupancy and leave memory underutilized. The paper’s motivation is clear and grounded in hardware profiling.
- PLoRA includes both a packing/scheduling component and an execution component with customized kernels, so the paper is not just proposing a scheduler in isolation; instead, it is reasonably complete.
- The empirical speedups are large and practically meaningful.
- The paper separates gains from the planner versus the optimized packed kernels and shows that both contribute materially, which strengthens the systems story.
- The method is tested on multiple model sizes and more than one model family, which is better than only evaluating on one narrow setup.

**Weaknesses**
- The paper’s main contribution is clearly systems-oriented, but the evaluation is narrower than the headline claims suggest. The experiments mostly involve a small set of NLP tasks and a synthetic-like search space of 120 LoRA configurations over only four tuning knobs.
- The main comparisons are Min GPU and Max GPU, which are fairly weak execution baselines. These are reasonable strawman baselines to include, but I would have liked stronger baselines such as alternative multi-job scheduling heuristics etc.
- The paper shows that the best adapter found in a 120-configuration search can outperform the default LoRA configuration, but that mostly demonstrates the value of broader search rather than proving that PLoRA itself improves model quality. In other words, the quality gains seem indirect, coming from being able to search more configurations faster.
- The cost model is important to the system, but I did not see enough detail in the main paper about how robust it is when extrapolating beyond the first few iterations or across more heterogeneous workloads.

---

> ### Author Rebuttal · Authors · 2026-03-31
>
> We thank the reviewer for their constructive and insightful feedback. Below, we address each weakness and question.
>
> W1: Evaluation scope.
>
> We conducted a thorough empirical study of hyperparameter sensitivity to establish a practical, useful design space for multi-LoRA training (Section 2.3).  We note that the 120 configurations are systematically constructed based on our sensitivity analysis (Appendix E.1), which shows the dominant factors affecting both model quality and resource usage. Concurrent works, such as [1] and [2],  generally fix an arbitrary number of LoRA adapters (4 in [1], 6 to 12 in [2]), whose scales are substantially smaller than those in our study. Therefore, we believe the scope and coverage of our empirical study already far exceed that of our concurrent works.
>
> W2 and Q1: Baselines.
>
> First, to the best of our knowledge, we are the first to investigate packing and scheduling in multi-LoRA training, enabling a new use case for concurrent LoRA training. Therefore, no head-to-head comparisons are available. For instance, existing works [1][2] generally fix the number of LoRA adapters. This can lead not only to underutilization but also to OOM in certain cases. We are the first to explicitly optimize this packing and scheduling decision, in conjunction with lower-level optimizations to support fast, concurrent training.
>
> Compared to concurrent works such as mLoRA and LoRAFusion, both of which target efficient kernels for pipeline parallelism but operate with a fixed number of adapters, PLoRA operates at one level above as an oracle scheduler, with the benefit of a powerful execution engine.
>
> If a user wants to train a set of LoRA adapters, PLoRA is the only option for breaking it into smaller, viable jobs. Once they have these jobs, they can then consider executing with the kernels provided by LoRAFusion or mLoRA if and only if they use pipeline parallelism (which is not widely supported in modern LoRA training libraries, such as [3][4][5]). Otherwise, they can still opt for PLoRA’s execution engine for widely supported TP and FSDP scenarios.
>
> To demonstrate the gains unlocked by adaptive scheduling, we compare the PLoRA scheduler with LoRAFusion kernels to a simple greedy scheduler that packs the default 4-adapter setup if possible, and halves it if OOM. Note that in order to accommodate LoRAFusion’s limitation on only supporting pipeline parallelism, we further restrict our evaluation to PP only. We follow a similar setup and evaluate Llama-3.1-8B and Qwen-2.5-32B across our LoRA config mix. We report the speedup in makespan in the following table
> | Makespan Reduction (↑)     | Llama-3.1-8B | Qwen-2.5-32B |
> |----------------------------|--------------|--------------|
> | LoRAFusion + PLoRA         | 2.62x        | 2.29x        |
> | LoRAFusion + Greedy        | 2.31x        | 2.05x        |
> | Min GPU                    | 1.00x        | 1.00x        |
>
> [1]. Zhu, Zhanda, et al. "LoRAFusion: Efficient LoRA Fine-Tuning for LLMs." EuroSys 2026.
>
> [2]. Lin, Sheng, et al. "LobRA: Multi-tenant Fine-tuning over Heterogeneous Data." VLDB 2025.
>
> [3]. Axolotl: Open Source LLM Post-Training. https://github.com/axolotl-ai-cloud/axolotl
>
> [4]. torchtune: PyTorch's finetuning library. https://github.com/meta-pytorch/torchtune
>
> [5]. Unsloth, Han et al. https://github.com/unslothai/unsloth
>
> W3 and Q3: Quality gains from configuration search are indirect.
>
> The reviewer is right that PLoRA is a systems contribution, not an algorithmic one for model quality. The model quality experiment (Section 4.4) is intended to demonstrate a practical benefit of PLoRA's efficiency: by reducing makespan, PLoRA enables users to explore more configurations within a fixed time/compute budget, thereby enabling them to find better-performing adapters. We do not claim that PLoRA directly improves the quality of any single LoRA run. The training procedure for each individual adapter is identical to standard LoRA fine-tuning. We will clarify this distinction more explicitly in the revision to avoid any misinterpretation.
>
> W4 and Q2: Insufficient detail on the cost model's robustness when extrapolating beyond the first few iterations or across heterogeneous workloads.
>
> The cost model (described in Appendix B) is an analytical model, not a learned predictor. It estimates memory usage and training time based on: (1) model architecture, LoRA configuration, and training strategies. Memory is computed analytically from parameter counts, gradient storage, and activation memory (Equations 19-20 in Appendix B). Training time is calibrated from the first few iterations of actual execution (10 iterations in our testbed). Since LoRA fine-tuning throughput is highly stable across iterations, extrapolation from early iterations is reliable. Across our experiments, the cost model's predictions are within 5% of actual values.

---

> > ### Author Rebuttal · Reviewer_Nteo · 2026-04-02
> >
> > All my concerns have been resolved and I would like to maintain my positive score.

---

### Official Review · Reviewer_DN1f · 2026-03-13

**Soundness:** 3
**Presentation:** 3
**Significance:** 2
**Originality:** 3
**Overall Recommendation:** 4
**Confidence:** 3

**Summary:**

This paper looks at how to make training lots of LoRA adapters more efficient. The authors point out that current methods don’t use hardware very well, especially when training multiple adapters at once. To tackle this, they introduce PLoRA—a system that lets you fine-tune several LoRA adapters at the same time. PLoRA uses a planner to organize things ahead of time and a special engine to run the training, along with custom code for running packed LoRA operations. Their results show it significantly speeds up training and shortens the time it takes, across different model sizes.

**Compliance With Llm Reviewing Policy:**

Affirmed.

**Final Justification:**

The rebuttal provides convincing evidence on both fronts I raised. In particular, the additional discussion of dynamic online workloads, along with the Poisson arrival experiment, demonstrates that PLoRA generalizes beyond static settings. The ablation results also clearly disentangle the contributions of the planner and custom kernels, providing a much clearer understanding of system level tradeoffs.

**Key Questions For Authors:**

How does PLoRA perform under more dynamic online workloads rather than mostly pre-specified configuration sets? What fraction of the gains comes from packing/planning versus custom packed kernels?

**Limitations:**

yes

**Strengths And Weaknesses:**

The motivation lands well: since LoRA is already widely adopted, figuring out how to train lots of adapters efficiently is a meaningful problem to tackle. It's also nice to see a paper address a real-world bottleneck instead of just proposing yet another modeling variant.

That said, the main tradeoff here is that the contribution leans heavily on systems engineering rather than algorithmic novelty. Which is totally fine for what this is—but it does mean the evaluation needs to be rock solid. I would've loved to see a bit more unpacking of the tradeoffs.

---

> ### Author Rebuttal · Authors · 2026-03-31
>
> We thank the reviewer for their constructive and insightful feedback. Below, we address each weakness and question.
>
> W1: Contribution leans heavily on systems engineering rather than algorithmic novelty; evaluation needs to be rock solid with more unpacking of tradeoffs.
>
> We appreciate this observation. As you rightly pointed out, system contributions are essential for making LoRA training practical at scale.
>
> We also appreciate the observation that this is a meaningful, real-world problem. We highlight the thoroughness of our evaluation below:
>
> We evaluate 6 models of 4 different sizes across two model families (Section 4.1, Figures 4a–4b) on a broadly constructed search space empirically supported by a thorough hyperparameter study (Section 2.3). In Appendix E.3, we also demonstrate how to integrate PLoRA with QLoRA, achieving 4.72× speedup over standard QLoRA fine-tuning on QWen-2.5-7B. We also run detailed ablations for different components of our system (Section 4.5 and Figure 6) and on kernels (Table 7-8).
>
> We believe this (in addition to other experiments not highlighted above) constitutes a comprehensive evaluation spanning multiple model families, model sizes, GPU types, quantization levels, and system components.
>
> Q1: How does PLoRA perform under more dynamic online workloads rather than mostly pre-specified configuration sets?
>
> PLoRA's two-stage design (offline planner + online execution engine) naturally accommodates dynamic scenarios. The Execution Engine (Section 3.1) includes a Resource Monitor that tracks GPU availability and a Job Launcher that dynamically dequeues and deploys jobs as resources become available. Thus, PLoRA natively handles a dynamic workload. When new LoRA configurations arrive at runtime, the Packing Planner queries the remaining configuration pool and can be re-invoked as new training jobs arrive.
>
> Regarding how to handle already running jobs, the hyperparameter tuning literature offers a standard approach: breaking a long-running job into smaller jobs to reduce bubbles arising from different training lengths [1]. This technique can also be applied in PLoRA, as checkpoints can be saved and reloaded in a future job.  In addition, PLoRA does not continue training completed adapters; each adapter tracks its own step count and stops when complete. The packed job continues only for the remaining active adapters.
>
>
> [1]. A system for massively parallel hyperparameter tuning. Li, et. al. MLSys 2020
>
> We demonstrate this in an experiment. For our workload, assume we start with 25% of the configurations available, with the remaining 75% arrive according to a Poisson distribution to maintain a dynamic workload. We still observe significant makespan reduction. Compared to a static workload, some configurations did not arrive in time, and therefore, some jobs are launched with lower utilization. However, this is only the property of the workload, not a drawback of PLoRA.
>
> | Makespan Reduction (↑)     | Llama-3.1-8B | Qwen-2.5-32B |
> |----------------------------|--------------|--------------|
> | Poisson(20) (PLoRA)        | 5.92x        | 5.80x        |
> | Poisson(20) (Min GPU)      | 1.00x        | 1.00x        |
>
> Q1: What fraction of the gains comes from packing/planning versus custom-packed kernels?
>
> As shown in our ablation study (Section 4.5, Figure 6), both components contribute materially. On QWen-2.5-3B and 7B, Sequential PLoRA (planner-only, no custom kernels) reduces the makespan by ~1.8x over Min GPU, while the full PLoRA system achieves up to 7.08x. The planner provides the foundation by enabling concurrent execution, while the custom kernels unlock the throughput potential of packed LoRA computations.

---

> > ### Author Rebuttal · Reviewer_DN1f · 2026-04-02
> >
> > The additional clarifications and experiments effectively address my main concerns. In particular, the discussion of dynamic online workloads, along with the Poisson arrival experiment, provides convincing evidence that PLoRA can handle realistic, non-static scenarios. The ablation results also clearly disentangle the contributions of the planner and custom kernels, which helps better understand the system-level tradeoffs.

---

### Decision · Program_Chairs · 2026-04-30

**Decision:**

Accept (regular)

**Comment:**

This submission presents PLORA, a system designed to enhance the efficiency of training multiple Low-Rank Adaptation (LoRA) adapters concurrently by addressing the underutilization of GPU resources in single-adapter jobs. The authors contribute a two-stage architecture featuring an offline packing planner that treats resource allocation as a combinatorial optimization problem and an online execution engine with specialized kernels, achieving significant improvement in training throughput. During the rebuttal, the authors successfully addressed concerns from two reviewers regarding dynamic workloads and system ablations through new Poisson arrival experiments and speedup disentanglement. However, Reviewer do8B remained concerned that the paper does not distinguish itself from serving systems because it primarily evaluates a fixed set of adapters, for which the AC noted that the authors added additional discussion of dynamic online workloads in rebuttal to Reviewer DN1f, which successfully demonstrated via the Poisson distribution experiment that the system maintains high efficiency even when jobs arrive non-deterministically.
Considering the overall consensus of the reviewers, the AC's final recommendation is a Weak Accept, with the strong suggestion that the authors incorporate their rebuttal clarifications and address remaining scalability concerns (sequence length ablation) and clarity on scope ( with hybrid parallelism strategies in large-scale settings) in the final manuscript.